# TEST-TIME RECALIBRATION OF CONFORMAL PREDICTORS UNDER DISTRIBUTION SHIFT BASED ON UNLABELED EXAMPLES

## ABSTRACT

Modern image classifiers achieve high predictive accuracy, but the predictions typically come without reliable uncertainty estimates. Conformal prediction algorithms provide uncertainty estimates by predicting a set of classes based on the probability estimates of the classifier (for example, the softmax scores). To provide such sets, conformal prediction algorithms often rely on estimating a cutoff threshold for the probability estimates, and this threshold is chosen based on a calibration set. Conformal prediction methods guarantee reliability only when the calibration set is from the same distribution as the test set. Therefore, the methods need to be recalibrated for new distributions. However, in practice, labeled data from new distributions is rarely available, making calibration infeasible. In this work, we consider the problem of predicting the cutoff threshold for a new distribution based on unlabeled examples only. While it is impossible in general to guarantee reliability when calibrating based on unlabeled examples, we show that our method provides excellent uncertainty estimates under natural distribution shifts, and provably works for a specific model of a distribution shift.

## 1 INTRODUCTION

Consider a (black-box) image classifier that is trained on a dataset to output probability estimates for $L$ classes given an input feature vector $\mathbf{x} \in \mathbb{R}^d$. This classifier is typically a deep neural network with a softmax layer at the end. Conformal prediction algorithms are wrapped around such a black-box classifier to generate a set of classes that contain the correct label with a user-specified desired probability based on the output probability estimates.

Let $\mathbf{x} \in \mathbb{R}^d$ be a feature vector with associated label $y \in \{1, \ldots, L\}$. We say that a set-valued function $\mathcal{C}$ generates valid prediction sets for the distribution $\mathcal{P}$ if

$$\mathrm{P}_{(\mathbf{x},y)\sim\mathcal{P}} \left[ y \in \mathcal{C}(\mathbf{x}) \right] \geq 1 - \alpha, \tag{1}$$

where $1 - \alpha$ is the desired coverage level. Conformal prediction methods generate valid set generating functions by utilizing a calibration set consisting of labeled examples $\{(\mathbf{x}_1, y_1), \ldots, (\mathbf{x}_n, y_n)\}$ drawn from the distribution $\mathcal{P}$. An important caveat of conformal prediction methods is that they assume that the examples from the calibration set and the test set are exchangeable, i.e., samples are identically distributed, or more broadly, are invariant to permutations across the two sets.

The exchangeability assumption is difficult to satisfy and verify in applications and potentially limits the applicability of conformal prediction methods in practice. In fact, in practice one usually expects a distribution shift between the calibration set and the examples at inference (or the test set), in which case the coverage guarantees provided by conformal prediction methods are void. For example, the new CIFAR-10.1 and ImageNetV2 test sets were created in the same way as the original CIFAR-10 and ImageNet test sets, yet Recht et al. (2019) found a notable drop in classification accuracy for all classifiers considered.

Ideally, a conformal predictor is recalibrated on a distribution before testing, otherwise the coverage guarantees are not valid (Cauchois et al., 2020). However, in real-world applications, while distribution shifts are ubiquitous, labeled data from new distributions is scarce or non-existent.

We therefore consider the problem of recalibrating a conformal predictor only based on unlabeled data from the new domain. This is an ill-posed problem: it is in general impossible to calibrate a conformal predictor based on unlabeled data. Yet, we propose a simple calibration method that gives excellent performance for a variety of natural distribution shifts.

**Organization and contributions.** We start with concrete examples on how conformal predictors yield miscalibrated uncertainty estimates under natural distribution shifts. We next propose a simple recalibration method that only uses unlabeled examples from the target distribution. We show that our method correctly recalibrates a popular conformal predictor (Sadinle et al., 2019) on a theoretical toy model. We provide empirical results for various natural distribution shifts of ImageNet showing that recalibrating conformal predictors using our proposed method significantly reduces the performance gap. In certain cases, it even achieves near oracle-level coverage.

**Related work.** Several works have considered robustness of conformal prediction to distribution shift. Tibshirani et al. (2019) considers covariate shifts and proposes calibrating conformal predictors by estimating the amount of covariate shift from unlabeled target data. Similarly, Park et al. (2022) considers the problem of estimating the covariate shift from unlabeled target data, and aims at constructing PAC prediction sets instead of the standard unconditionally valid prediction sets. In contrast, we focus on complex image datasets for which covariate shift is not well defined. In Appendix B, we provide a comparison of our method to the above covariate shift based methods for a setting where we have access to labeled examples from multiple domains during training/calibration, one of which correspond to the target distribution.

We are not aware of other works studying calibration of conformal predictors under distribution shift based on unlabeled examples. However, prior works propose to make conformal predictors robust to various distribution shifts from the source distribution of the calibration set (Cauchois et al., 2020; Gendler et al., 2022), via calibrating the conformal predictor to achieve a desired coverage in the worse case scenario of the considered distribution shifts. Cauchois et al. (2020) considers covariate shifts and calibrates the conformal predictor to achieve coverage for the worst-case distribution within the $f$-divergence ball of the source distribution. Gendler et al. (2022) considers adversarial perturbations as distribution shifts and calibrates a conformal predictor to achieve coverage for the worst-case distribution obtained through $\ell_2$-norm bounded adversarial noise.

While making the conformal predictor robust to a range of worst-case distributions at calibration time allows maintaining coverage worst-case distributions, this approaches has two shortcomings: 1. Natural distribution shifts are difficult to capture mathematically, and models like covariate-shifts or adversarial perturbations do not seem to model natural distribution shifts (such as that from ImageNet to ImageNetV2) accurately. 2. Calibrating for a worst-case scenario results in an overly conservative conformal predictor that tends to yield much higher coverage than desired for test distributions that correspond to a less severe shift from the source, which comes at the cost of reduced efficiency (i.e., larger set size, or larger confidence interval length).

In contrast, our method does not compromise the efficiency of the conformal predictor on easier distributions as we recalibrate the conformal predictor separately for any new dataset.

A related problem is to predict the accuracy of a classifier on new distributions from unlabeled data sampled from the new distribution (Deng & Zheng, 2021; Chen et al., 2021; Jiang et al., 2021; Deng et al., 2021; Guillory et al., 2021; Garg et al., 2022). In particular, Garg et al. (2022) proposed a simple method that achieves state-of-the-art performance in predicting classifier accuracy across a range of distributions. However, the calibration problem we consider is fundamentally different than estimating the accuracy of a classifier. While predicting the accuracy of the classifier would allow making informed decisions on whether to use the classifier for a new distribution, it doesn't provide a solution to recalibrate.

## 2 CONFORMAL PREDICTION AND PROBLEM STATEMENT

We start by introducing conformal prediction and our problem setup.

**Conformal prediction.** Consider a black-box classifier with input feature vector $\mathbf{x} \in \mathbb{R}^d$ that outputs a probability estimate $\pi_\ell(\mathbf{x}) \in [0, 1]$ for each class $\ell = 1, \dots, L$. Typically, the classifier is a

neural network trained on some distribution, and the probability estimates are the softmax outputs. We denote the order statistics of the probability estimates by $\pi_{(1)}(\mathbf{x}) \geq \pi_{(2)}(\mathbf{x}) \geq \ldots \geq \pi_{(L)}(\mathbf{x})$.

Many conformal predictors are based on calibrating on a calibration set $\mathcal{D}_{\text{cal}}^{\mathcal{P}} = \{(\mathbf{x}_i, y_i)\}_{i=1}^n$ to find a cutoff threshold (Sadinle et al., 2019; Romano et al., 2020; Angelopoulos et al., 2020) that achieves the desired empirical coverage on this set. Here, the superscript $\mathcal{P}$ denotes the distribution from which the examples in the calibration set are sampled from. Specifically, conformal calibration computes the threshold parameter as

$$\tau^* = \inf \left\{ \tau : |\{i : y_i \in \mathcal{C}(\mathbf{x}_i, u_i, \tau)\}| \geq (1 - \alpha)(n + 1) \right\}. \tag{2}$$

Here, $\mathcal{C}(\mathbf{x}, u, \tau)$ is the set-valued function associated with the conformal predictor that outputs a confidence set which is a subset of the classes $\{1, \ldots, L\}$, and $u_i$ is added randomization to ensure the smoothness of the cardinality term, chosen independently and uniformly from the interval $[0, 1]$. See Vovk et al. (2005) on smoothed conformal predictors. Finally, the '+1' term in the $(n + 1)$ term is a bias correction for the finite size of the calibration set.

This conformal calibration procedure achieves distributional coverage as defined in the expression (1), for any set valued function $\mathcal{C}(\mathbf{x}, u, \tau)$ satisfying the nesting property, i.e., $\mathcal{C}(\mathbf{x}, u, \tau_1) \subseteq \mathcal{C}(\mathbf{x}, u, \tau_2)$ for $\tau_1 < \tau_2$, see (Angelopoulos et al., 2020, Thm. 1).

In this paper, we primarily focus on the popular conformal predictors *Thresholded Prediction Sets* (TPS) (Sadinle et al., 2019) and *Adaptive Prediction Sets* (APS) Romano et al. (2020). The set generating functions of the two conformal predictors are

$$\mathcal{C}^{\text{TPS}}(\mathbf{x}, \tau) = \{\ell \in \{1, \ldots, L\} : \pi_\ell(\mathbf{x}) \geq 1 - \tau\}, \tag{3}$$

and

$$\mathcal{C}^{\text{APS}}(\mathbf{x}, u, \tau) = \left\{ \ell \in \{1, \ldots, L\} : \sum_{j=1}^{\ell-1} \pi_{(j)}(\mathbf{x}) + u \cdot \pi_{(\ell)}(\mathbf{x}) \leq \tau \right\}, \tag{4}$$

with $u \sim U(0, 1)$ for smoothing. The set generating function of TPS doesn't require smoothing since each softmax score is independently thresholded and therefore there are no discrete jumps.

We emphasize that the threshold $\tau$ depends on the distribution $\mathcal{P}$ and a labeled calibration set from $\mathcal{P}$ is needed to compute the threshold through conformal calibration given in the expression (2). We therefore add a superscript to the threshold to make clear which distribution was used to sample the calibration set, for example $\tau^{\mathcal{P}}$ indicates that the calibration set was sampled from the distribution $\mathcal{P}$. The prediction set function $\mathcal{C}^{\text{TPS}}$ for TPS and $\mathcal{C}^{\text{APS}}$ for APS both satisfy the nesting property. Therefore, TPS/APS calibrated on a calibration set $\mathcal{D}_{\text{cal}}^{\mathcal{P}}$ by computing the threshold in the expression (2) is guaranteed to achieve coverage on the distribution $\mathcal{P}$. However, coverage is only guaranteed if the test distribution $\mathcal{Q}$ is the same as the calibration distribution $\mathcal{P}$.

**Coverage failures under distributions shifts.** Often we're most interested in quantifying uncertainty with conformal predictions when we apply a classifier to new data that might come from a slightly different distribution than the distribution we calibrated on. Yet, conformal predictors only provide coverage guarantees for data coming from the same distribution as the calibration set, and the coverage guarantees often fail even under slight distribution shifts.

For example, APS calibrated to yield $1 - \alpha = 0.9$ coverage on the standard ImageNet-Val dataset only achieves a coverage of $0.64$ on the ImageNet-Sketch dataset, which consists of sketches of the ImageNet-Val images and hence constitutes a distribution shift (Wang et al., 2019).

Different conformal predictors typically have different coverage gaps under the same distribution shift. More efficient conformal predictors (i.e., those that produce smaller prediction sets) tend to have a larger coverage gap under a distribution shift. For example, both TPS and RAPS (a generalization of APS proposed by Angelopoulos et al. (2020)) yield smaller confidence sets, but only achieve a coverage of $0.38$ vs. $0.64$ for APS on the ImageNet-Sketch distribution shift discussed above.

Even under more subtle distribution shifts such as a subpopulation shifts (Santurkar et al., 2021), the achieved coverage can drop significantly. For example, APS calibrated to yield $1 - \alpha = 0.9$ coverage on the source distribution of the Living-17 BREEDS dataset only achieves a coverage of $0.68$ on the target distribution. The source and target distributions contain images of exclusively different breeds of animals while the animals' species is shared as the label (Santurkar et al., 2021).

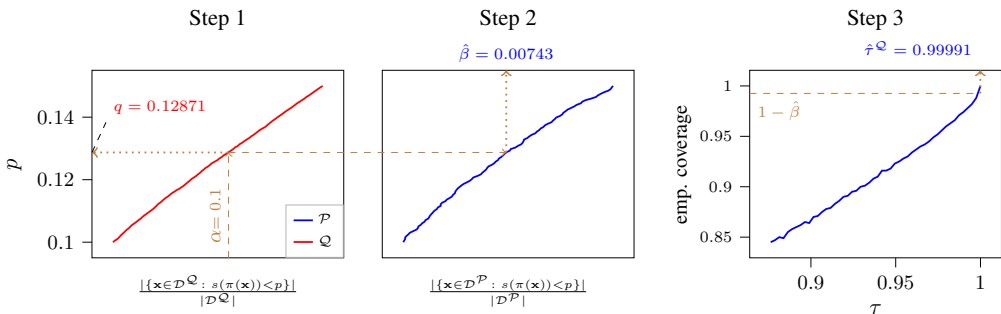

Figure 1: Illustration of recalibration of a conformal predictor with QTC: ImageNet-Val was used as the source distribution $\mathcal{P}$ and ImageNet-Sketch as the target distribution $\mathcal{Q}$.
**Step 1:** QTC is used on the target to find the threshold $q$ such that a $\alpha$ fraction of the target samples have a lower score than $q$ as measured by $\frac{|\{\mathbf{x} \in \mathcal{D}^{\mathcal{Q}} : s(\pi(\mathbf{x})) < q\}|}{|\mathcal{D}^{\mathcal{Q}}|} = \alpha$.
**Step 2:** At test time, QTC estimates the value of $\beta$ as $\frac{|\{\mathbf{x} \in \mathcal{D}^{\mathcal{P}} : s(\pi(\mathbf{x})) < q\}|}{|\mathcal{D}^{\mathcal{P}}|}$.
**Step 3:** The estimate of $\beta$ is used to recalibrate the conformal predictor, yielding $\hat{\tau}^{\mathcal{Q}}_{\text{QTC}} = 0.99991$.

**Problem statement.** Our goal is to recalibrate a conformal predictor on a new distribution $\mathcal{Q}$ based on only unlabeled data. Specifically, our goal is to predict the threshold $\tau^{\mathcal{Q}}$ as an estimate $\hat{\tau}^{\mathcal{Q}}$ for a target distribution $\mathcal{Q}$ as well as possible so that the conformal predictor with the set function $\mathcal{C}(\mathbf{x}, u, \hat{\tau}^{\mathcal{Q}})$ achieves coverage close to the desired coverage of $1 - \alpha$ on the new distribution $\mathcal{Q}$. Towards this goal, we only have access to an unlabeled dataset $\mathcal{D}^{\mathcal{Q}}_{\text{unlabeled}} = \{\mathbf{x}_1, \ldots, \mathbf{x}_n\}$ sampled from the target distribution $\mathcal{Q}$. We use $\mathcal{D}_{\text{unlabeled}}$ to denote an unlabeled (label-stripped) dataset.

In general, it is impossible to guarantee coverage since conformal prediction relies on strict exchangeability assumptions which can not be controlled in practice for new datasets (Vovk et al., 2005; Romano et al., 2020; Angelopoulos et al., 2020; Cauchois et al., 2020). However, we will see that we can consistently estimate the threshold $\tau^{\mathcal{Q}}$ for a variety of natural distribution shifts.

We refer to the difference between the target coverage of $1 - \alpha$ and the actual coverage achieved on a given distribution without any recalibration efforts as the *coverage gap*. We assess how effective a recalibration method is based on the reduction of the coverage gap after recalibration under a variety of settings (e.g., choice of conformal predictor, target level $1 - \alpha$) and distribution shifts.

## 3 METHODS

In this section we introduce our calibration method, that we term Quantile Thresholded Confidence (QTC), along with baseline methods we consider in our experiments.

### 3.1 QUANTILE THRESHOLDED CONFIDENCE

Consider a conformal predictor calibrated on the source distribution $\mathcal{P}$ to yield coverage of $1 - \alpha$, which yields the threshold $\tau^{\mathcal{P}}_{\alpha}$. The conformal predictor guarantees coverage of $1 - \alpha$ on the source distribution $\mathcal{P}$, but on a different distribution $\mathcal{Q}$ the coverage is off. But there is a value $\beta$ such that, if we calibrate the conformal predictor on the *source distribution* using the value $\beta$ instead of $\alpha$, it achieves $1 - \alpha$ coverage on the *target distribution*, i.e., the corresponding thresholds obey $\tau^{\mathcal{P}}_{\beta} = \tau^{\mathcal{Q}}_{\alpha}$.

Our method estimates the value $\beta$ based on unlabeled examples. From this estimate $\hat{\beta}$, we can compute the corresponding threshold $\tau^{\mathcal{P}}_{\hat{\beta}}$ by calibrating the conformal predictor on the source calibration set using $\hat{\beta}$, which yields a threshold close to the desired one, i.e., $\tau^{\mathcal{P}}_{\hat{\beta}} \approx \tau^{\mathcal{Q}}_{\alpha}$.

We are given a labeled source dataset $\mathcal{D}_{\text{cal}}^{\mathcal{P}}$ and an unlabeled target dataset $\mathcal{D}_{\text{unlabeled}}^{\mathcal{Q}}$. Our estimate of $\beta$ relies on the following quantile function

$$q(\mathcal{D}, c) = \inf \left\{ p \colon \frac{1}{|\mathcal{D}|} \sum_{\mathbf{x} \in \mathcal{D}} \mathbb{1}_{\{s(\pi(\mathbf{x})) < p\}} \geq c \right\}, \tag{5}$$

which depends on the scores of the classifier through a score function $s(\pi(\mathbf{x})) = \max_\ell \pi_\ell(\mathbf{x})$, which we take as the largest softmax score of the classifier's prediction. Here, $\mathcal{D}$ is a set of unlabeled examples and $c \in [0, 1]$ is a scalar. Our method first identifies a threshold based on the unlabeled target dataset $\mathcal{D}_{\text{unlabeled}}^{\mathcal{Q}}$ for a desired coverage level $\alpha$ in expression (5) by computing $q(\mathcal{D}_{\text{unlabeled}}^{\mathcal{Q}}, \alpha)$. Since this process is identical to finding the $(\alpha)^{th}$ quantile of the scores on the dataset, we dub the method Quantile Thresholded Confidence (QTC). QTC estimates $\beta$ as

$$\beta_{\text{QTC}}(\mathcal{D}_{\text{unlabeled}}^{\mathcal{Q}}) = \frac{1}{|\mathcal{D}_{\text{cal}}^{\mathcal{P}}|} \sum_{\mathbf{x} \in \mathcal{D}_{\text{cal}}^{\mathcal{P}}} \mathbb{1}_{\left\{ s(\pi(\mathbf{x})) < q(\mathcal{D}_{\text{unlabeled}}^{\mathcal{Q}}, \alpha) \right\}}. \tag{6}$$

We use the value $\beta_{\text{QTC}}$ found in (6) to calibrate the conformal predictor on the dataset $\mathcal{D}_{\text{cal}}^{\mathcal{P}}$ with the goal of achieving $1 - \alpha$ coverage on the target distribution $\mathcal{Q}$. QTC is illustrated in Figure 1.

We also study two variants of QTC that find the threshold $q$ on the source dataset used for conformal calibration instead of the target as ablation methods: QTC-SC and QTC-ST, where SC stands for Source Coverage and ST for Source Threshold. Similarly to QTC, QTC-SC aims to estimate the value $\beta$. On the other hand, QTC-ST aims to directly estimate the threshold $\tau^{\mathcal{Q}}$ by using the threshold $\tau^{\mathcal{P}}$ found by calibrating the conformal predictor on the source as in equation (2). QTC-SC and QTC-ST estimate $\beta$ and $\tau^{\mathcal{Q}}$ as

$$\beta_{\text{QTC-SC}}(\mathcal{D}) = 1 - \frac{1}{|\mathcal{D}|} \sum_{\mathbf{x} \in \mathcal{D}} \mathbb{1}_{\left\{ s(\pi(\mathbf{x})) < q(\mathcal{D}_{\text{cal}}^{\mathcal{P}}, 1-\alpha) \right\}}, \tag{7}$$

$$\tau_{\text{QTC-ST}}(\mathcal{D}) = \frac{1}{|\mathcal{D}|} \sum_{\mathbf{x} \in \mathcal{D}} \mathbb{1}_{\left\{ s(\pi(\mathbf{x})) < q(\mathcal{D}_{\text{cal}}^{\mathcal{P}}, \tau^{\mathcal{P}}) \right\}}, \tag{8}$$

where $\mathcal{D}_{\text{unlabeled}}^{\mathcal{Q}}$ is substituted for $\mathcal{D}$ in the above expressions for target distribution $\mathcal{Q}$.

QTC is inspired by a method for predicting the accuracy of a classifier from Garg et al. (2022). Garg et al. (2022)'s method finds a threshold on the scores that matches the accuracy of a classifier on the dataset and predicts the accuracy on other datasets analogous to the expression (8). Contrary, we predict a threshold of a conformal predictor, and our method is based on predicting the parameter $\beta$ instead of a threshold directly.

## 3.2 BASELINE METHODS

We consider several regression-based methods as baselines. Regression-based methods have been used for predicting classification accuracy, assuming a correlation between the classification accuracy and a feature (e.g., average confidence) across different distributions (Deng et al., 2021; Deng & Zheng, 2021; Guillory et al., 2021). Those regression-based methods learn a function $f_{\boldsymbol{\theta}}$ that makes a prediction on a new distribution $\mathcal{Q}$ based on the set of unlabeled examples $\mathcal{D}_{\text{unlabeled}}^{\mathcal{Q}}$. The function $f_{\boldsymbol{\theta}}$ is learned based on synthetically generated distributions $\mathcal{P}_1, \mathcal{P}_2, \ldots$ and datasets $\mathcal{D}_{\text{cal}}^{\mathcal{P}_1}, \mathcal{D}_{\text{cal}}^{\mathcal{P}_2}, \ldots$ sampled from those distributions. Such synthetic datasets can be produced by applying corruptions to the natural images of the source distribution, e.g., ImageNet-C (Hendrycks & Dietterich, 2019).

We expand the existing range of regression-based methods by considering a variety of setups for the feature space that extract different levels of detail from the classifier's statistics on the dataset. We use $\phi_\pi(\mathcal{D}) \colon \mathbb{R}^L \to \mathbb{R}^d$ to denote the feature extractor that maps the softmax scores of the classifier to the features for a given dataset $\mathcal{D}$. A simple example is the one-dimensional feature ($d = 1$) extracted by computing the average confidence of a given classifier across the examples of a given dataset.

We fit a regression function $f_\theta$ parameterized by a the feature extractor $\phi_\pi$ by minimizing the mean squared error between the output and the calibrated threshold $\tau$ across the distributions as

$$\hat{\theta} = \arg\min_\theta \sum_j (f_\theta(\phi_\pi(\mathcal{D}_j)) - \tau^{\mathcal{P}_j})^2. \tag{9}$$

We consider the following choices for the feature extractor $\phi_\pi$:

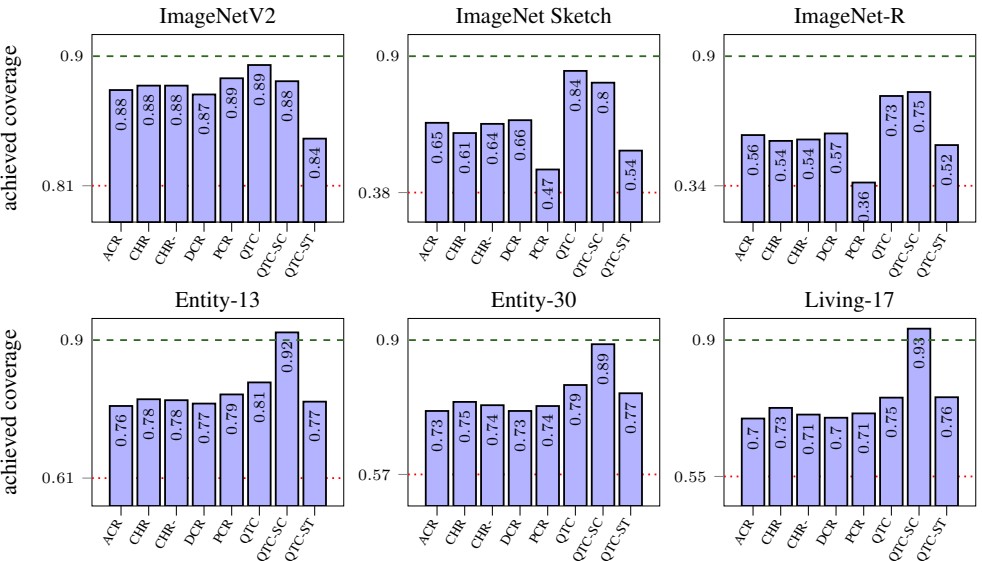

Figure 2: Coverage obtained by TPS for a desired coverage of $1 - \alpha = 0.9$ on the target distribution $\mathcal{Q}$ after recalibration using the unlabeled samples from $\mathcal{Q}$ for various recalibration methods. The dotted line is the coverage without recalibration, and the dashed line is the target coverage $1 - \alpha = 0.9$. The figure shows that QTC and its variants almost fully close the coverage gap across ImageNet and BREEDS test distribution shifts, corresponding to varying severities.

- *Average confidence regression (ACR)*: The average confidence of the classifier across the entire dataset which is $\phi_\pi(\mathcal{D}) = \frac{1}{|\mathcal{D}|} \sum_{\mathbf{x} \in \mathcal{D}} \max_\ell \pi_\ell(\mathbf{x})$.

- *Difference of confidence regression (DCR)* (Guillory et al., 2021): The average confidence of the classifier across the entire dataset offset by the average confidence on the source dataset, which is $\phi_\pi(\mathcal{D}) = \frac{1}{|\mathcal{D}|} \sum_{\mathbf{x} \in \mathcal{D}} \max_\ell \pi_\ell(\mathbf{x}) - \frac{1}{|\mathcal{D}^{\mathcal{P}}|} \sum_{\mathbf{x} \in \mathcal{D}^{\mathcal{P}}} \max_\ell \pi_\ell(\mathbf{x})$, where $\mathcal{D}^{\mathcal{P}}$ is the source dataset. Prediction is also for the offset target $\tau - \tau^{\mathcal{P}}$.

  We consider DCR in addition to ACR, because DCR performs better for predicting the classifier accuracy (Guillory et al., 2021). Since the threshold $\tau$ found by conformal calibration depends on the distribution of the confidences beyond the average, we propose the below techniques for extracting more detailed information from the dataset.

- *Confidence histogram-density regression (CHR)*: Variable dimensional ($d = p$) features extracted as $\phi_\pi(\mathcal{D}) = \left\{ \frac{1}{|\mathcal{D}|} \sum_{\mathbf{x} \in \mathcal{D}} \mathbb{1}_{\left\{ \max_\ell \pi_\ell(\mathbf{x}) \in \left[ \frac{j-1}{p}, \frac{j}{p} \right] \right\}} \right\}_{j = \{1, \dots, p\}}$. This corresponds to the normalized histogram density of the classifier confidence across the dataset, where p is a hyperparameter that determines the number of histogram bins in the probability range $[0, 1]$. Neural networks tend to be overconfident in their prediction which heavily skews the histogram densities to the last bin. We also therefore consider a variant of CHR, *dubbed CHR-*, where we have $j = \{1, \dots, p-1\}$ and hence $d = p - 1$, equivalent to dropping the last bin of the histogram as a feature.

- *Predicted class-wise average confidence regression (PCR)*: Features with dimensionality equal to the number of classes ($d = L$) extracted as $\phi_\pi(\mathcal{D}) = \left\{ \frac{\sum_{\mathbf{x} \in \mathcal{D}} \pi_j(\mathbf{x}) \cdot \mathbb{1}_{\{l = \arg \max_\ell \pi_\ell(\mathbf{x})\}}}{\sum_{\mathbf{x} \in \mathcal{D}} \mathbb{1}_{\{l = \arg \max_\ell \pi_\ell(\mathbf{x})\}}} \right\}_{j = \{1, \dots, L\}}$. This corresponds to the average confidence of the classifier across the samples for each predicted class.

## 4 EXPERIMENTAL RESULTS

For most experiments, we use ResNet-50 and DenseNet-121 as the backbone classifiers. We use the following choices for the source distribution $\mathcal{P}$ and associated natural distribution shifts:

**ImageNet (Deng et al., 2009) distribution shifts:** ImageNet is one of the most popular image classification datasets. In our ImageNet experiments, we take ImageNet as the source distribution $\mathcal{P}$ and we consider the following natural distribution shifts of ImageNet as the target distribution $\mathcal{Q}$:

- **ImageNetV2** (Recht et al., 2019) was constructed by following the same procedure as for constructing and labeling the original ImageNet dataset. However, all standard models perform significantly worse on ImageNetV2 relative to the original ImageNet testset.
- **ImageNet-Sketch** (Wang et al., 2019) contains sketch-like images of the objects in the original ImageNet, but otherwise matches the original categories and scales.
- **ImageNet-R** (Hendrycks et al., 2021) contains artwork images of the ImageNet class objects found in the web. ImageNet-R only contains images for a 200-class subset of the original ImageNet. We don't limit our experiments to this subset but instead consider the adverse setting of calibrating on all 1000 classes since our main goal is to provide an end-to-end solution for recalibration of the conformal predictors and we are interested in how well our method performs against possible adversaries such as dataset imbalance that can be encountered in practice.

For ImageNet and the above variations, we use a fixed ResNet-50 and DenseNet-121 that were pretrained on the ImageNet training set.

**BREEDS (Santurkar et al., 2021) distribution shifts:** The BREEDS datasets were constructed using the existing ImageNet images, but use different classes. BREEDS utilizes the hierarchical WordNet structure of the classes to choose a parent class that makes the original ImageNet classes the leaves. For example, in the BREEDS Living-17 dataset, one of the classes is *domestic cat*. This is a parent class of several ImageNet classes, which are *tiger cat, Egyptian cat, Persian cat and Siamese cat*. BREEDS induces a subpopulation shift from the source distribution to the target by assigning these leaf classes exclusively to either the source or target. For example, the images in the source dataset of Living-17 under the *domestic cat* class are that of either *tiger cats* or *Egyptian cats*, whereas in the target are that of either *Persian cats* or *Siamese cats*. Therefore, despite having the same label (*domestic cat*), the source and target distributions semantically differ due to the differences between the breeds, which induces a subpopulation shift. We consider three BREEDS datasets: Entity-13, Entity-30 and Living-17, which are named using the convention *theme/object type–#classes*. Since we can't use ImageNet pre-trained models (even with finetuning, as all layers have been trained on all ImageNet classes already), we train a ResNet-18 model from scratch for the BREEDS datasets such that the classifiers only sees examples from the source distribution.

For all experiments, we first calibrate the conformal predictor on the source distribution $\mathcal{P}$ to find the cutoff threshold $\tau^{\mathcal{P}}$. For QTC and variants, we find the threshold $q$ using the expression (5). For the regression methods, we use the ImageNet-C dataset (Hendrycks & Dietterich, 2019) as the source of synthetic distributions, find the cutoff threshold $\tau$ for each of the distributions, and fit a regressor by minimizing the loss (9). For the regression function we use a 4-layer MLP with ReLU activations. ImageNet-C consists of 90 different distributions obtained by synthetically perturbing the images of ImageNet-Val for 18 different types of perturbations at 5 different levels of severity, resulting in 90 distinct distributions.

**Recalibration experiments for a fixed target coverage.** We first evaluate the recalibration methods for a fixed target coverage of $1 - \alpha = 0.9$. The results in Figure 2 for recalibrating TPS show that while regression methods can fall short on reducing the coverage gap, QTC is able to reduce the coverage gap significantly and even close it in many cases. Often QTC works best, but for the distribution shifts of BREEDS, QTC-SC performs better. This can be attributed to the difference of the nature of shifts between ImageNet distributions and BREEDS distribution.

**Recalibration experiments for different target coverage levels.** The coverage gap (i.e., the difference of achieved coverages and targeted coverage) varies across the desired coverage level $1 - \alpha$. We there next evaluate the performance as a function of the desired coverage level. Figure 3 shows the coverage obtained after recalibration TPS and APS for different values of $1 - \alpha$ for the natural distribution shifts from ImageNet. Both QTC and QTC-SC close the coverage gap significantly for all choices of $1 - \alpha$, whereas the best performing regression-based baseline method, CHR-, fails to consistently significantly improve the coverage gap across all choices of $1 - \alpha$.

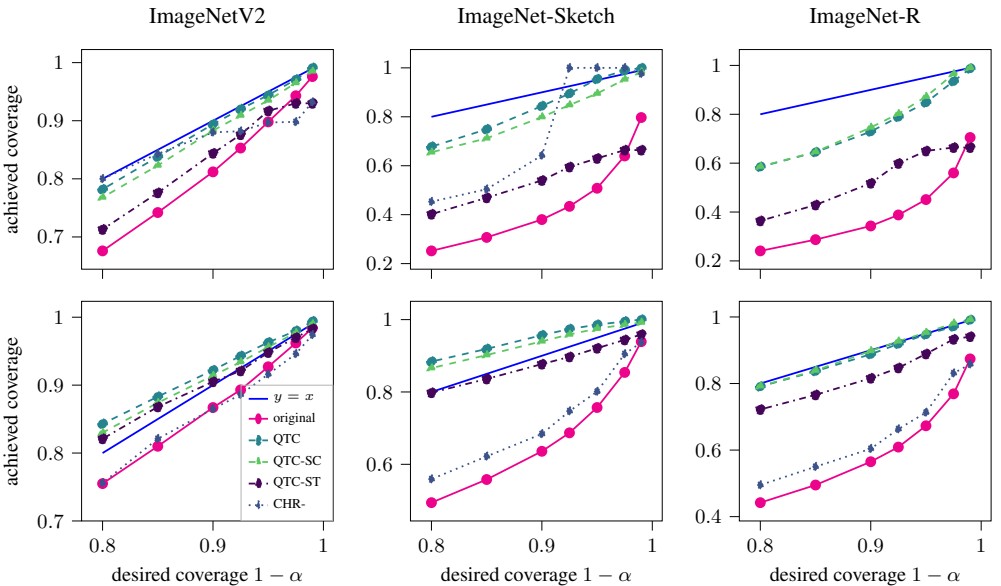

Figure 3: Coverage obtained by TPS (**top row**) and APS (**bottom row**) on the target distribution $\mathcal{Q}$ for various settings of $(1 - \alpha)$ after recalibration with the respective prediction method. For regression methods, only the best performing method of CHR- is shown. Both QTC and QTC-SC significantly close the coverage gap across the range of $1 - \alpha$, while CHR- yields inconsistent or insufficient performance improvements.

## 5 THEORETICAL RESULTS

We consider a simple binary classification distribution shift model from Nagarajan et al. (2021); Garg et al. (2022), and adapt the analysis from Garg et al. (2022) to show that recalibrating provably succeeds within this model. Specifically, we show that the conformal predictor TPS with QTC yields the desired coverage of $1 - \alpha$ on the target distribution based on unlabeled examples.

We start by describing the distribution shift model from Garg et al. (2022). Consider a binary classification problem with response $y \in \{-1, 1\}$ and with two features $\mathbf{x} = [x_{\text{inv}}, x_{\text{sp}}] \in \mathbb{R}^2$, an invariant one and a spuriously correlated one. The source and target distributions $\mathcal{P}$ and $\mathcal{Q}$ over the feature vector and label are defined as follows. The label $y$ is uniform distributed over $\{-1, 1\}$. The invariant fully-predictive feature $x_{\text{inv}}$ is uniformly distributed in an interval determined by the constants $c > \gamma \geq 0$, with the interval being conditional on $y$:

$$x_{\text{inv}}|y \sim \begin{cases} U[\gamma, c] & \text{if} \quad y = 1 \\ U[-c, -\gamma] & \text{if} \quad y = -1 \end{cases}. \tag{10}$$

The spurious feature $x_{\text{sp}}$ is correlated with the response $y$ such that $\mathrm{P}_{(\mathbf{x},y)\sim\mathcal{P}}[x_{\text{sp}} \cdot y > 0] = p^{\mathcal{P}}$, where $p^{\mathcal{P}} \in (0.5, 1.0)$ for some joint distribution $\mathcal{P}$. A distribution shift is modeled by simulating target data with different degrees of spurious correlation such that $\mathrm{P}_{(\mathbf{x},y)\sim\mathcal{Q}}[x_{\text{sp}} \cdot y > 0] = p^{\mathcal{Q}}$, where $p^{\mathcal{Q}} \in [0, 1]$. There is a distribution shift from source to target when $p^{\mathcal{P}} \neq p^{\mathcal{Q}}$. Two example distributions $\mathcal{P}$ and $\mathcal{Q}$ are illustrated in Figure 4.

We consider a logistic regression classifier that predicts class probability estimates for the classes $y = -1$ and $y = 1$ as $f(\mathbf{x}) = \left[\frac{1}{1+e^{\mathbf{w}^T \mathbf{x}}}, \frac{e^{\mathbf{w}^T \mathbf{x}}}{1+e^{\mathbf{w}^T \mathbf{x}}}\right]$, where $\mathbf{w} = [w_{\text{inv}}, w_{\text{sp}}] \in \mathbb{R}^2$. Note that the classifier with $w_{\text{inv}} > 0$ and $w_{\text{sp}} = 0$ minimizes the misclassification error across all choices of distributions $\mathcal{P}$ and $\mathcal{Q}$ (i.e., across all choices of $p$). However, a classifier learned by minimizing the empirical logistic loss via gradient descent depends on both the invariant feature $x_{\text{inv}}$ and the spuriously-correlated feature $x_{\text{sp}}$, i.e., $w_{\text{sp}} \neq 0$ due to the geometric skews on the finite data and statistical skews of the optimization with finite gradient descent steps (Nagarajan et al., 2021).

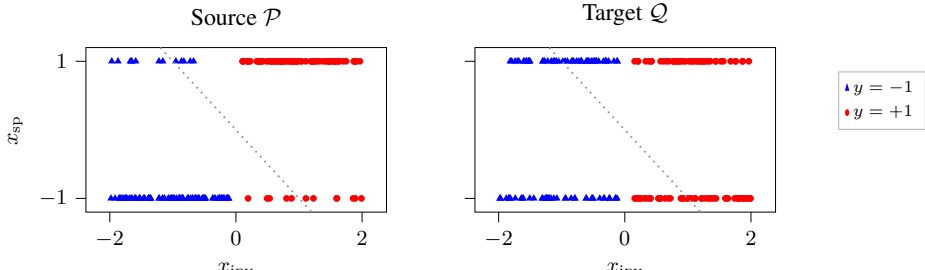

Figure 4: Example source and target distributions $\mathcal{P}$ and $\mathcal{Q}$ for the binary classification model, and a classifier with $w_{\mathrm{inv}}, w_{\mathrm{sp}} = 1$. The decision boundary is shown with a faded dotted line. The correlation between the feature $x_{\mathrm{sp}}$ and the label $y$ is higher for the source than target ($p^{\mathcal{P}} > p^{\mathcal{Q}}$).

We consider the conformal predictor TPS (Sadinle et al., 2019) applied to this problem to generate confidence sets. We extend the results of Garg et al. (2022, Theorem 3) and show that for a logistic regression classifier described as above (i.e., $w_{\mathrm{inv}} > 0, w_{\mathrm{sp}} \neq 0$), TPS recalibrated using QTC provably yields $1 - \alpha$ coverage on the target distribution:

**Theorem 1** (Informal). *Consider the logistic regression classifier for the binary classification problem described above with $w_{\mathrm{inv}} > 0, w_{\mathrm{sp}} \neq 0$, let $n$ be the number of samples for the source and target datasets and $\alpha \in (0, \epsilon)$ be a user-defined value, where $\epsilon$ is the error rate of the classifier on the source. The coverage achieved on the target by recalibrating TPS on the source with the QTC estimate obtained in* (6) *by finding the QTC threshold on the target as in* (5) *converges to $1 - \alpha$ as $n \to \infty$ with high probability.*

First, we comment on the assumption on $\alpha$. A value of $\alpha$ that is larger than the error rate of the classifier would not make sense as it would result in empty confidence sets for a portion of the examples in the dataset. This is because returning a confidence set consisting of the top prediction alone always achieves a coverage of $1 - \epsilon$ by definition.

In order to understand the intuition behind Theorem 1, we first explain how the coverages is off under a distribution shift in this model. Consider a classifier that depends positively on the spurious feature (i.e., $w_{\mathrm{sp}} > 0$). When the spurious correlation is decreased from the source distribution to the target, the error rate of the classifier increases. TPS calibrated on the source samples finds a threshold $\tau$ such that the prediction sets yield $1 - \alpha$ coverage on the source dataset as $n \to \infty$. In other words, the fraction of misclassified points for which the model confidence is larger than the threshold $\tau$ is equal to $\alpha$ on the source. As the spurious correlation decreases and the error rate increases from source to target, the fraction of misclassified points for which the model confidence is larger than the threshold $\tau$ surpasses $\alpha$, leading to a gap in targeted and actual coverage.

Now, we remark on how QTC recalibrates and ensures the target coverage is meet. Note that there exists an unknown coverage $1 - \beta$ that can be used to calibrate TPS on the source distribution such that it yields $1 - \alpha$ coverage on the target. Theorem 1 guarantees that QTC correctly estimates $\beta$ and therefore recalibration of the conformal predictor using QTC yields the desired coverage level of $1 - \alpha$ on the target.

## 6 CONCLUSION

We considered the problem of providing reliable uncertainty estimates for conformal prediction algorithms under distribution shifts based on unlabeled examples. We propose a simple test-time recalibration method dubbed Quantile Thresholded Confidence (QTC) that recalibrates conformal predictors based only on unlabeled examples. QTC provably succeeds on the distribution shift model from Nagarajan et al. (2021); Garg et al. (2022), and most importantly reduces, or even closes, the coverage gap (i.e., the difference of achieved coverage and desired coverage) of conformal predictors under distribution shifts for a variety of natural distribution shifts.

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

# A    PROOF OF THEOREM 1

In this section, we state and prove a formal version of Theorem 1. Our results rely on adapting the proof idea of Garg et al. (2022, Theorem 3) for predicting the classification accuracy of a model to our conformal prediction setup.

Recall that we consider a distribution shift model for a binary classification problem with an invariant predictive feature and a spuriously correlated feature, where a distribution shift is induced by the spurious feature of the target distribution being more or less correlated with the label than the source distribution (Nagarajan et al., 2021; Garg et al., 2022).

We consider a logistic regression classifier that outputs class probability estimates for the two classes of $y = -1$ and $y = +1$ as

$$\mathbf{f}(\mathbf{x}) = \left[ \frac{1}{1 + e^{\mathbf{w}^T \mathbf{x}}}, \frac{e^{\mathbf{w}^T \mathbf{x}}}{1 + e^{\mathbf{w}^T \mathbf{x}}} \right],$$

where $\mathbf{w} = [w_{\text{inv}}, w_{\text{sp}}] \in \mathbb{R}^2$. The classifier with $w_{\text{inv}} > 0$ and $w_{\text{sp}} = 0$ minimizes the misclassification error across all choices of distributions $\mathcal{P}$ and $\mathcal{Q}$ (i.e., across all choices of $p$). However, a classifier learned by minimizing the empirical logistic loss via gradient descent depends on both the invariant feature $x_{\text{inv}}$ and the spuriously-correlated feature $x_{\text{sp}}$, i.e., $w_{\text{sp}} \neq 0$ due to the geometric skews on the finite data and statistical skews of the optimization with finite gradient descent steps (Nagarajan et al., 2021).

In order to understand how geometric skews result in learning a classifier that depends on the spurious feature, suppose the probability that the spurious feature agrees with the label is high, i.e., $p$ is close to 1.0. Note that in a finite-size training set drawn from this distribution, the fraction of samples for which the spurious feature disagrees with the label (i.e., $x_{\text{sp}} \neq y$) is small. Therefore, the margin on the invariant feature for these samples alone can be significantly larger than the actual margin $\gamma$ of the underlying distribution. This implies that the max-margin classifier depends positively on the spurious feature, i.e., $w_{\text{sp}} > 0$. Furthermore, we assume that $w_{\text{inv}} > 0$, which is required to obtain non-trivial performance (beating a random guess).

**Conformal prediction in the distribution shift model.**    We consider the conformal prediction method TPS (Sadinle et al., 2019) applied to the linear classifier described above. While other conformal prediction methods such as APS and RAPS also work for this model, the smoothing induced by the randomization of the model scores used in those conformal predictors would introduce additional complexity to the analysis. TPS also tends to be more efficient in that it yields smaller confidence sets compared to APS and RAPS at the same coverage level, see (Angelopoulos et al., 2020, Table 9).

In the remaining part of this section, we establish Theorem 1, which states that TPS recalibrated on the source calibration set with QTC achieves the desired coverage of $1 - \alpha$ on any target distribution that has a (potentially) different correlation probability $p$ for the spurious feature. We show this in two steps:

First, we show that the oracle TPS that achieves $1 - \alpha$ coverage on the target distribution achieves $1 - \beta$ coverage on the source distribution. We then show that TPS calibrated to achieve $1 - \beta$ empirical coverage on the source dataset achieves at least $1 - \alpha$ coverage on the target distribution, as $n \to \infty$.

Second, we provide a bound on the deviation of the QTC estimate from the true value of $\beta$. We show that in the infinite sample size case, the QTC estimate converges to the true value of $\beta$. Those two steps prove Theorem 1.

Recall that the calibration of TPS entails identifying a cutoff threshold $\tau$ computed by the formula (2). The set generating function of TPS for the linear classification problem described above simplifies to

$$\mathcal{C}^{\text{TPS}}(\mathbf{x}, \tau) = \{\ell \in \{0, 1\} \colon f_\ell(\mathbf{x}) \geq 1 - \tau\}, \tag{11}$$

where $f_0(\mathbf{x})$ and $f_1(\mathbf{x})$ are the first and second entry of $\mathbf{f}(\mathbf{x})$ as defined above.

We are only interested in the regime where the desired coverage level $1 - \alpha$ is larger than the classifier's accuracy, or equivalently $\alpha < \epsilon$ with $\epsilon$ being the error rate of the classifier. This is because

a trivial method that constructs confidence sets with equal length of 1 for all samples (i.e., singleton sets of only the predicted label) already achieves coverage of $1 - \epsilon$.

**Step 1:** QTC relies on the fact that there exists an unknown $\beta \in (0, 1)$ that can be used to calibrate TPS on the source distribution such that it yields $1 - \alpha$ coverage on the target. Here, we show that TPS calibrated to achieve $1 - \beta$ coverage on the source calibration set $\mathcal{D}_{\mathrm{cal}}^{\mathcal{P}}$ via computing the threshold (2) achieves $1 - \alpha$ coverage on the target distribution $\mathcal{Q}$ as $n \to \infty$.

We utilize the following coverage guarantee of conformal predictors established by Vovk et al. (2005); Lei et al. (2017); Angelopoulos et al. (2020):

**Lemma 1.** *(Lei et al., 2017, Thm. 2.2), (Angelopoulos et al., 2020, Thm. 1, Prop. 1) Consider $(\mathbf{x}_i, y_i), i = 1, \ldots, n$ drawn iid from some distribution $\mathcal{P}$. Let $\mathcal{C}(\mathbf{x}, \tau)$ be the conformal set generating function that satisfies the nesting property in $\tau$, i.e., $\mathcal{C}(\mathbf{x}, \tau') \subseteq \mathcal{C}(\mathbf{x}, \tau)$ if $\tau' \leq \tau$. Then, the conformal predictor calibrated by finding $\tau^*$ that achieves $1 - \alpha$ coverage on the finite set $\{(\mathbf{x}_i, y)\}_{i=1}^n$ as in (2) satisfies the coverage over a sample $(\mathbf{x}_{n+1}, y_{n+1})$ drawn from the same distribution $\mathcal{P}$*

$$\mathrm{P}\left[y_{n+1} \in \mathcal{C}(\mathbf{x}_{n+1}, \tau^*)\right] \geq 1 - \alpha. \tag{12}$$

*Furthermore, assume that the variables $s_i = s(\mathbf{x}_i, y_i) = \inf\{\tau : y_i \in \mathcal{C}(\mathbf{x}_i, \tau)\}$ for $i = 1, \ldots, n$ are distinct almost surely. Then, the coverage achieved by the calibrated conformal predictor with the set generating function $\mathcal{C}(\mathbf{x}, \tau) = \{\ell \in \mathcal{Y} : s(\mathbf{x}, \ell) \leq \tau\}$ is also accurate, in that it satisfies*

$$\mathrm{P}\left[y_{n+1} \in \mathcal{C}(\mathbf{x}_{n+1}, \tau^*)\right] \leq 1 - \alpha + \frac{1}{n+1}. \tag{13}$$

Both the lower bound (12) and the upper bound (13) of Lemma 1 apply to TPS in the context of the binary classification problem that we consider. To see this, we verify that TPS calibrated with the set generating function (11) satisfies both assumptions of Lemma 1. First, note that TPS satisfies the nesting property, since we have $\mathcal{C}^{\mathrm{TPS}}(\mathbf{x}, \tau') \subseteq \mathcal{C}^{\mathrm{TPS}}(\mathbf{x}, \tau)$ for $\tau' \leq \tau$. Next, note that for TPS we have $s(\mathbf{x}, y) = f_y(\mathbf{x})$. Further note that the linear logistic regression model we consider assigns a distinct score to each data point and since the invariant feature $x_{\mathrm{inv}}$ is uniformly distributed in a continuous interval conditional on $y$, the variables $s_i$ are distinct almost surely.

Now, consider the oracle TPS threshold $\tau_\alpha^{\mathcal{Q}}$ that achieves $1 - \alpha$ coverage, or equivalently $\alpha$ miscoverage, on the target distribution, i.e.,

$$\mathrm{P}_{(\mathbf{x}, y) \sim \mathcal{Q}}\left[y \notin \mathcal{C}^{\mathrm{TPS}}(\mathbf{x}, \tau_\alpha^{\mathcal{Q}})\right] = \alpha. \tag{14}$$

Next, note that $y \notin \mathcal{C}^{\mathrm{TPS}}(\mathbf{x}, \tau_\alpha^{\mathcal{Q}})$ if and only if $\arg\max_{j \in \{0,1\}} f_j(\mathbf{x}) \neq y$ *and* $\max_{j \in \{0,1\}} f_j(\mathbf{x}) \geq \tau_\alpha^{\mathcal{Q}}$. To see that, note that the confidence set returned by TPS is a singleton containing only the top prediction of the model when the confidence of this prediction is higher than the threshold $\tau_\alpha^{\mathcal{Q}}$. Moreover, the confidence set returned by TPS for the binary classification problem above does not contain the true label only when the confidence set is the singleton set of the top prediction of the model and is different than the true label. Thus, equation (14) implies

$$\mathrm{P}_{(\mathbf{x}, y) \sim \mathcal{Q}}\left[\arg\max_{j \in \{0,1\}} f_j(\mathbf{x}) \neq y \text{ and } \max_{j \in \{0,1\}} f_j(\mathbf{x}) \geq \tau_\alpha^{\mathcal{Q}}\right] = \alpha. \tag{15}$$

We define $\beta$ as the miscoverage that the oracle TPS yields on the source distribution, i.e.,

$$\beta := \mathrm{P}_{(\mathbf{x}, y) \sim \mathcal{P}}\left[\arg\max_{j \in \{0,1\}} f_j(\mathbf{x}) \neq y \text{ and } \max_{j \in \{0,1\}} f_j(\mathbf{x}) \geq \tau_\alpha^{\mathcal{Q}}\right]. \tag{16}$$

We have $\beta \neq \alpha$ if there is a distribution shift from target to source.

Consider the threshold $\hat{\tau}_\beta^{\mathcal{P}}$ found by calibrating TPS on the set $\mathcal{D}_{\mathrm{cal}}^{\mathcal{P}}$ to achieve empirical coverage of $1 - \beta$ as in (2). TPS with the threshold $\hat{\tau}_\beta^{\mathcal{P}}$ achieves coverage on the source distribution $\mathcal{P}$ as a result of Lemma 1. Moreover, combining (12) with (13) at $n \to \infty$ yields exact coverage of $1 - \beta$ on the source distribution $\mathcal{P}$. Thus, we have

$$\mathrm{P}_{(\mathbf{x}, y) \sim \mathcal{P}}\left[\arg\max_{j \in \{0,1\}} f_j(\mathbf{x}) \neq y \text{ and } \max_{j \in \{0,1\}} f_j(\mathbf{x}) \geq \hat{\tau}_\beta^{\mathcal{P}}\right] = \beta. \tag{17}$$

Comparing equation (17) to the definition of $\beta$ in equation (16) yields $\hat{\tau}_\beta^{\mathcal{P}} = \tau_\alpha^{\mathcal{Q}}$. Therefore, it follows that TPS calibrated to achieve $1 - \beta$ coverage on the source calibration set $\mathcal{D}_{\mathrm{cal}}^{\mathcal{P}}$ as in (2) achieves exactly $1 - \alpha$ coverage on the target distribution $\mathcal{Q}$ as $n \to \infty$.

**Step 2:** In the second step, we show that QTC correctly estimates the value of $\beta$ described above. This is formalized in the theorem below.

**Theorem 2.** *Given the logistic regression classifier for the binary classification problem described above with any $w_{\mathrm{inv}} > 0, w_{\mathrm{sp}} \neq 0$, assume that the threshold $q$ for QTC is computed using a dataset $\mathcal{D}^{\mathcal{Q}}$ consisting of $n$ samples, sampled from some target distribution $\mathcal{Q}$, such that*

$$\frac{1}{n} \sum_{\mathbf{x} \in \mathcal{D}^{\mathcal{Q}}} \mathbb{1}_{\left\{\max_{j \in \{0,1\}} f_j(\mathbf{x}) < q\right\}} = \alpha. \tag{18}$$

*Consider the oracle TPS conformal predictor with conformal threshold $\tau_\alpha^{\mathcal{Q}}$, i.e., the predictor that achieves $1 - \alpha$ coverage on the target distribution $\mathcal{Q}$. Denote with $1 - \beta$ the coverage achieved on the source distribution $\mathcal{P}$ by this oracle TPS. Fix a $\delta > 0$. The QTC estimate of the miscoverage $\beta$, denoted by*

$$\beta_{\mathrm{QTC}} = \frac{1}{|\mathcal{D}_{\mathrm{cal}}^{\mathcal{P}}|} \sum_{\mathbf{x} \in \mathcal{D}_{\mathrm{cal}}^{\mathcal{P}}} \mathbb{1}_{\{s(\pi(\mathbf{x})) < q\}}, \tag{19}$$

*satisfies the following inequality with probability at least $1 - \delta$ over a randomly drawn set of examples $\mathcal{D}^{\mathcal{Q}}$*

$$|\beta_{\mathrm{QTC}} - \beta| \leq \sqrt{\frac{2 \log(16/\delta)}{n \cdot c_{sp}}}, \tag{20}$$

*where $c_{\mathrm{sp}} = (1 - p^{\mathcal{Q}}) \cdot (1 - p^{\mathcal{P}})^2$ if $w_{\mathrm{sp}} > 0$ and $c_{\mathrm{sp}} = p^{\mathcal{Q}} \cdot (p^{\mathcal{P}})^2$ otherwise.*

*Proof.* We adapt the proof idea of Garg et al. (2022, Theorem 3), which pertains to the problem of estimating the classification error of the classifier on the target, to estimating the source coverage of the oracle conformal predictor that achieves $1 - \alpha$ coverage on the target distribution.

For notational convenience, we define the event that a sample $(\mathbf{x}, y)$ is not in the prediction set of the oracle TPS with conformal threshold $\tau_\alpha^{\mathcal{Q}}$ (i.e., $y \notin \mathcal{C}^{\mathrm{TPS}}(\mathbf{x}, \tau_\alpha^{\mathcal{Q}})$) as

$$\mathcal{E}_{mc} = \{y \notin \mathcal{C}^{\mathrm{TPS}}(\mathbf{x}, \tau_\alpha^{\mathcal{Q}})\}$$
$$= \{\arg \max_{j \in \{0,1\}} f_j(\mathbf{x}) \neq y \text{ and } \max_{j \in \{0,1\}} f_j(\mathbf{x}) \geq \tau_\alpha^{\mathcal{Q}}\}.$$

**The infinite sample size case ($\mathbf{n} \to \infty$).** In this part we show that as $n \to \infty$, the QTC estimate $\beta_{\mathrm{QTC}}$ found as in (19) converges to the source miscoverage $\beta$ to illustrate the proof idea. For $n \to \infty$, the QTC estimate $\beta_{\mathrm{QTC}}$ in (19) becomes

$$\beta_{\mathrm{QTC}} = \mathbb{E}_{(\mathbf{x}, y) \sim \mathcal{P}} \left[ \mathbb{1}_{\left\{\max_{j \in \{0,1\}} f_j(\mathbf{x}) \leq q\right\}} \right]$$
$$= \mathrm{P}_{(\mathbf{x}, y) \sim \mathcal{P}} \left[ \max_{j \in \{0,1\}} f_j(\mathbf{x}) < q \right]$$
$$= \mathrm{P}_{(\mathbf{x}, y) \sim \mathcal{P}} [\mathcal{E}_{mc}] \tag{21}$$
$$= \beta,$$

where the last equality is the definition of $\beta$ as given in equation (16). The critical step is equation (21), which we establish in the remainder of this part of the proof.

First, we condition on the label $y$. Using the law of total probability, we get

$$\mathrm{P}_{(\mathbf{x}, y) \sim \mathcal{P}} \left[ \max_{j \in \{0,1\}} f_j(\mathbf{x}) < q \right] = \mathrm{P}_{\mathbf{x} \sim \mathcal{P}|y=-1} \left[ \max_{j \in \{0,1\}} f_j(\mathbf{x}) < q \right] \cdot \mathrm{P}_{(\mathbf{x}, y) \sim \mathcal{P}} [y = -1]$$
$$+ \mathrm{P}_{\mathbf{x} \sim \mathcal{P}|y=+1} \left[ \max_{j \in \{0,1\}} f_j(\mathbf{x}) < q \right] \cdot \mathrm{P}_{(\mathbf{x}, y) \sim \mathcal{P}} [y = +1]$$
$$\stackrel{(i)}{=} \frac{1}{2} \cdot \mathrm{P}_{\mathbf{x} \sim \mathcal{P}|y=-1} \left[ \max_{j \in \{0,1\}} f_j(\mathbf{x}) < q \right]$$
$$+ \frac{1}{2} \cdot \mathrm{P}_{\mathbf{x} \sim \mathcal{P}|y=+1} \left[ \max_{j \in \{0,1\}} f_j(\mathbf{x}) < q \right]$$
$$\stackrel{(ii)}{=} \mathrm{P}_{\mathbf{x} \sim \mathcal{P}|y} \left[ \max_{j \in \{0,1\}} f_j(\mathbf{x}) < q \right]. \tag{22}$$

For equation $(i)$, we used that $y$ is uniformly distributed across $\{-1, 1\}$, and for equation $(ii)$ that $\mathbf{x}$ is symmetrically distributed with respect to the label $y$. That is, we have $x_{\text{inv}} \sim U[-c, -\gamma]$ and $\mathrm{P}\left[x_{\text{sp}} = -1\right] = p$ if $y = -1$ and $x_{\text{inv}} \sim U[\gamma, c]$ and $\mathrm{P}\left[x_{\text{sp}} = +1\right] = p$ if $y = +1$, so the two probabilities in $(i)$ are equal.

We can further expand the expression for the probability $\mathrm{P}_{\mathbf{x} \sim \mathcal{P}|y}\left[\max_{j \in \{0,1\}} f_j(\mathbf{x}) < q\right]$ by additionally conditioning on the spurious feature $x_{\text{sp}}$, which yields

$$\mathrm{P}_{(\mathbf{x},y) \sim \mathcal{P}}\left[\max_{j \in \{0,1\}} f_j(\mathbf{x}) < q\right] = \mathrm{P}_{x_{\text{inv}} \sim \mathcal{P}|y, x_{\text{sp}}=y}\left[\max_{j \in \{0,1\}} f_j(\mathbf{x}) < q\right] \cdot \mathrm{P}_{\mathbf{x} \sim \mathcal{P}|y}\left[x_{\text{sp}} = y\right]$$
$$+ \mathrm{P}_{x_{\text{inv}} \sim \mathcal{P}|x_{\text{sp}} \neq y}\left[\max_{j \in \{0,1\}} f_j(\mathbf{x}) < q\right] \cdot \mathrm{P}_{\mathbf{x} \sim \mathcal{P}|y}\left[x_{\text{sp}} \neq y\right].$$
(23)

In order to simplify the expression in the RHS of equation (23), we consider the cases of $w_{\text{sp}} > 0$ and $w_{\text{sp}} < 0$ separately. If $w_{\text{sp}} > 0$, we have $\max_{j \in \{0,1\}} f_j(\mathbf{x}) > q$ if $x_{\text{sp}} = y$. Therefore, we have $\mathrm{P}_{x_{\text{inv}} \sim \mathcal{P}|y, x_{\text{sp}}=y}\left[\max_{j \in \{0,1\}} f_j(\mathbf{x}) < q\right] = 0$ if $w_{\text{sp}} > 0$ and equation (23) simplifies to

$$\mathrm{P}_{(\mathbf{x},y) \sim \mathcal{P}}\left[\max_{j \in \{0,1\}} f_j(\mathbf{x}) < q\right] = \mathrm{P}_{x_{\text{inv}} \sim \mathcal{P}|x_{\text{sp}} \neq y}\left[\max_{j \in \{0,1\}} f_j(\mathbf{x}) < q\right] \cdot \mathrm{P}_{\mathbf{x} \sim \mathcal{P}|y}\left[x_{\text{sp}} \neq y\right]$$
$$= \mathrm{P}_{x_{\text{inv}} \sim \mathcal{P}|x_{\text{sp}} \neq y}\left[\max_{j \in \{0,1\}} f_j(\mathbf{x}) < q\right] \cdot (1 - p^{\mathcal{P}}).$$
(24)

Similarly, if $w_{\text{sp}} < 0$, we have $\max_{j \in \{0,1\}} f_j(\mathbf{x}) > q$ if $x_{\text{sp}} \neq y$, and equation (23) becomes

$$\mathrm{P}_{(\mathbf{x},y) \sim \mathcal{P}}\left[\max_{j \in \{0,1\}} f_j(\mathbf{x}) < q\right] = \mathrm{P}_{x_{\text{inv}} \sim \mathcal{P}|x_{\text{sp}}=y}\left[\max_{j \in \{0,1\}} f_j(\mathbf{x}) < q\right] \cdot \mathrm{P}_{\mathbf{x} \sim \mathcal{P}|y}\left[x_{\text{sp}} = y\right]$$
$$= \mathrm{P}_{x_{\text{inv}} \sim \mathcal{P}|x_{\text{sp}}=y}\left[\max_{j \in \{0,1\}} f_j(\mathbf{x}) < q\right] \cdot p^{\mathcal{P}}.$$
(25)

We next follow the same steps that we carried out above for $\mathrm{P}_{(\mathbf{x},y) \sim \mathcal{P}}\left[\max_{j \in \{0,1\}} f_j(\mathbf{x}) < q\right]$ to rewrite the probability $\mathrm{P}_{(\mathbf{x},y) \sim \mathcal{P}}\left[\mathcal{E}_{mc}\right]$. If $w_{\text{sp}} > 0$, the classifier makes no errors if $x_{\text{sp}} = y$ and only misclassifies a fraction of examples if $x_{\text{sp}} \neq y$. Therefore, we have

$$\mathrm{P}_{\mathbf{x} \sim \mathcal{P}|y}\left[\mathcal{E}_{mc}\right] = \mathrm{P}_{x_{\text{inv}} \sim \mathcal{P}|x_{\text{sp}} \neq y}\left[\mathcal{E}_{mc}\right] \cdot \mathrm{P}_{\mathbf{x} \sim \mathcal{P}|y}\left[x_{\text{sp}} \neq y\right]$$
$$= \mathrm{P}_{x_{\text{inv}} \sim \mathcal{P}|x_{\text{sp}} \neq y}\left[\mathcal{E}_{mc}\right] \cdot (1 - p^{\mathcal{P}}).$$
(26)

Similarly, for $w_{\text{sp}} < 0$, we have

$$\mathrm{P}_{\mathbf{x} \sim \mathcal{P}|y}\left[\mathcal{E}_{mc}\right] = \mathrm{P}_{x_{\text{inv}} \sim \mathcal{P}|x_{\text{sp}} \neq y}\left[\mathcal{E}_{mc}\right] \cdot \mathrm{P}_{\mathbf{x} \sim \mathcal{P}|y}\left[x_{\text{sp}} = y\right]$$
$$= \mathrm{P}_{x_{\text{inv}} \sim \mathcal{P}|x_{\text{sp}} \neq y}\left[\mathcal{E}_{mc}\right] \cdot p^{\mathcal{P}}.$$
(27)

Therefore, in order to establish equation (21), it suffices to show

$$\mathrm{P}_{x_{\text{inv}} \sim \mathcal{P}|y, x_{\text{sp}} \neq y}\left[\max_{j \in \{0,1\}} f_j(\mathbf{x}) < q\right] = \mathrm{P}_{x_{\text{inv}} \sim \mathcal{P}|y, x_{\text{sp}} \neq y}\left[\mathcal{E}_{mc}\right], \quad \text{for } w_{\text{sp}} > 0 \text{ and}$$
(28)

$$\mathrm{P}_{x_{\text{inv}} \sim \mathcal{P}|y, x_{\text{sp}}=y}\left[\max_{j \in \{0,1\}} f_j(\mathbf{x}) < q\right] = \mathrm{P}_{x_{\text{inv}} \sim \mathcal{P}|y, x_{\text{sp}}=y}\left[\mathcal{E}_{mc}\right], \quad \text{for } w_{\text{sp}} < 0.$$
(29)

The feature $x_{\text{inv}}$ is identically distributed conditioned on $y$, i.e., uniformly distributed in the same interval, regardless of the underlying source or target distributions $\mathcal{P}$ and $\mathcal{Q}$. Therefore, equations (28) and (29) are equivalent to

$$\mathrm{P}_{x_{\text{inv}} \sim \mathcal{Q}|y, x_{\text{sp}} \neq y}\left[\max_{j \in \{0,1\}} f_j(\mathbf{x}) < q\right] = \mathrm{P}_{x_{\text{inv}} \sim \mathcal{Q}|y, x_{\text{sp}} \neq y}\left[\mathcal{E}_{mc}\right], \quad \text{for } w_{\text{sp}} > 0 \text{ and}$$
(30)

$$\mathrm{P}_{x_{\text{inv}} \sim \mathcal{Q}|y, x_{\text{sp}}=y}\left[\max_{j \in \{0,1\}} f_j(\mathbf{x}) < q\right] = \mathrm{P}_{x_{\text{inv}} \sim \mathcal{Q}|y, x_{\text{sp}}=y}\left[\mathcal{E}_{mc}\right], \quad \text{for } w_{\text{sp}} < 0.$$
(31)

Equations (30) and (31) in turn follow from

$$\mathrm{P}_{(\mathbf{x},y)\sim\mathcal{Q}}\left[\max_{j\in\{0,1\}}f_j(\mathbf{x})<q\right]=\mathrm{P}_{(\mathbf{x},y)\sim\mathcal{Q}}\left[\mathcal{E}_{mc}\right],\tag{32}$$

by carrying out the same steps that we carried out to expand the probabilities $\mathrm{P}_{\mathbf{x}\sim\mathcal{P}|y}\left[\mathcal{E}_{mc}\right]$ and $\mathrm{P}_{(\mathbf{x},y)\sim\mathcal{P}}\left[\mathcal{E}_{mc}\right]$ starting from equation (32) to establish equations (28) and (29). Equation (32) in turn is a consequence of combining (15) with (18) at $n\to\infty$. This establishes equation (21), as desired.

**The finite sample case:** We next show that the desired results approximately hold with high probability over a randomly drawn finite-sized set of examples $\mathcal{D}^{\mathcal{Q}}$. We bound the difference between the LHS and RHS of (30) and (31) with high probability.

First, consider the case of $w_{\mathrm{sp}}>0$. Recall that for the case of $w_{\mathrm{sp}}>0$ we are interested in the regime where $w_{\mathrm{sp}}\neq y$. We denote the set of points in the target set $\mathcal{D}^{\mathcal{Q}}$ for which the spurious feature disagrees with the label as

$$\mathcal{X}_D=\{i=1,\dots,n:x_{\mathrm{sp}}\neq y,(\mathbf{x}_i,y_i)\in\mathcal{D}^{\mathcal{Q}}\},$$

and denote the set of points for which the spurious feature agrees with the label as

$$\mathcal{X}_A=\{i=1,\dots,n:x_{\mathrm{sp}}=y,(\mathbf{x}_i,y_i)\in\mathcal{D}^{\mathcal{Q}}\}.$$

Note that the QTC threshold $q$ found on the entire set $\mathcal{D}^{\mathcal{Q}}$ as in (18) satisfies

$$\frac{1}{|\mathcal{X}_D|}\sum_{i\in\mathcal{X}_D}\mathbb{1}_{\left\{\max_{j\in\{0,1\}}f_j(\mathbf{x}_i)<q\right\}}=\frac{1}{|\mathcal{X}_D|}\sum_{i\in\mathcal{X}_D}\mathbb{1}_{\{\mathcal{E}_{mc}(\mathbf{x}_i,y_i)\}},\tag{33}$$

which follows from noting that the classifier only makes an error on the subset $\mathcal{X}_D$ if $w_{\mathrm{sp}}>0$ and therefore the only points for which the event $\mathcal{E}_{mc}$ is observed lie in the set $\mathcal{X}_D$. Similarly, as established before in the infinite sample case, we have $\mathbb{1}_{\left\{\max_{j\in\{0,1\}}f_j(\mathbf{x}_i)<q\right\}}=0$ for all $i\in\mathcal{X}_D$.

By the Dvoretzky-Kiefer-Wolfowitz-Massart (DKWM) inequality, for any $q>0$ we have with probability at least $1-\delta/8$

$$\left|\frac{1}{|\mathcal{X}_D|}\sum_{i\in\mathcal{X}_D}\mathbb{1}_{\left\{\max_{j\in\{0,1\}}f_j(\mathbf{x}_i)<q\right\}}-\mathbb{E}_{x_{\mathrm{inv}}\sim\mathcal{Q}|y,x_{\mathrm{sp}}\neq y}\left[\mathbb{1}_{\left\{\max_{j\in\{0,1\}}f_j(\mathbf{x})<q\right\}}\right]\right|\leq\sqrt{\frac{\log(16/\delta)}{2|\mathcal{X}_D|}}.\tag{34}$$

Plugging equation (33) into (34), we have with probability at least $1-\delta/8$

$$\left|\mathbb{E}_{x_{\mathrm{inv}}\sim\mathcal{Q}|y,x_{\mathrm{sp}}\neq y}\left[\mathbb{1}_{\left\{\max_{j\in\{0,1\}}f_j(\mathbf{x})<q\right\}}\right]-\frac{1}{|\mathcal{X}_D|}\sum_{i\in\mathcal{X}_D}\mathbb{1}_{\{\mathcal{E}_{mc}\}}\right|\leq\sqrt{\frac{\log(16/\delta)}{2|\mathcal{X}_D|}}.\tag{35}$$

We next bound the second term in the LHS of equation (35) from its expectation. Using Hoeffding's inequality, we have with probability at least $1-\delta/8$

$$\left|\frac{1}{|\mathcal{X}_D|}\sum_{i\in\mathcal{X}_D}\mathbb{1}_{\{\mathcal{E}_{mc}\}}-\mathbb{E}_{x_{\mathrm{inv}}\sim\mathcal{Q}|y,x_{\mathrm{sp}}\neq y}\left[\mathbb{1}_{\{\mathcal{E}_{mc}\}}\right]\right|\leq\sqrt{\frac{\log(16/\delta)}{2|\mathcal{X}_D|}}.\tag{36}$$

Combining equations (35) and (36) using the triange inequality and union bound, we have with probability at least $1-\delta/4$

$$\left|\mathbb{E}_{x_{\mathrm{inv}}\sim\mathcal{Q}|y,x_{\mathrm{sp}}\neq y}\left[\mathbb{1}_{\left\{\max_{j\in\{0,1\}}f_j(\mathbf{x})<q\right\}}\right]-\mathbb{E}_{x_{\mathrm{inv}}\sim\mathcal{Q}|y,x_{\mathrm{sp}}\neq y}\left[\mathbb{1}_{\{\mathcal{E}_{mc}\}}\right]\right|\leq\sqrt{\frac{2\log(16/\delta)}{|\mathcal{X}_D|}}.\tag{37}$$

Recall that the invariant feature $x_{\mathrm{inv}}$ is uniformly distributed in the same interval conditioned on $y$ regardless of the source or target distributions $\mathcal{P}$ and $\mathcal{Q}$ and that $\mathrm{P}_{x_{\mathrm{inv}}|y,x_{\mathrm{sp}}=y}\left[\max_{j\in\{0,1\}}f_j(\mathbf{x})>q\right]=\mathrm{P}_{x_{\mathrm{inv}}|y,x_{\mathrm{sp}}=y}\left[\arg\max_{j\in\{0,1\}}f_j(\mathbf{x})\neq y\right]=0$ for the

case of $w_{sp} > 0$ as shown before. Therefore, by dividing both sides of (37) with $1/\mathrm{P}_{\mathbf{x}\sim\mathcal{P}|y}[x_{sp} \neq y]$ we have with probability at least $1 - \delta/4$

$$\left|\mathbb{E}_{(\mathbf{x},y)\sim\mathcal{P}}\left[\mathbb{1}_{\left\{\max_{j\in\{0,1\}} f_j(\mathbf{x})<q\right\}}\right] - \mathbb{E}_{(\mathbf{x},y)\sim\mathcal{P}}\left[\mathbb{1}_{\{\mathcal{E}_{mc}\}}\right]\right| \leq \frac{1}{\mathrm{P}_{\mathbf{x}\sim\mathcal{P}|y}[x_{sp} \neq y]}\sqrt{\frac{2\log(16/\delta)}{|\mathcal{X}_D|}}$$
$$= \frac{1}{1-p^{\mathcal{P}}}\sqrt{\frac{2\log(16/\delta)}{|\mathcal{X}_D|}}.$$
(38)

For the case of $w_{sp} < 0$, we can show an analogous result by noting that the above results can be shown on the set $\mathcal{X}_A$, where $x_{sp} = y$. Specifically, noting that $\frac{1}{|\mathcal{X}_A|}\sum_{i\in\mathcal{X}_A}\mathbb{1}_{\left\{\max_{j\in\{0,1\}} f_j(\mathbf{x}_i)<q\right\}} = \frac{1}{|\mathcal{X}_A|}\sum_{i\in\mathcal{X}_A}\mathbb{1}_{\{\mathcal{E}_{mc}\}}$ if $w_{sp} < 0$ and following exactly the same steps from equation (33) onward that lead to equation (38), we have with probability at least $1 - \delta/4$

$$\left|\mathbb{E}_{(\mathbf{x},y)\sim\mathcal{P}}\left[\mathbb{1}_{\left\{\max_{j\in\{0,1\}} f_j(\mathbf{x})<q\right\}}\right] - \mathbb{E}_{(\mathbf{x},y)\sim\mathcal{P}}\left[\mathbb{1}_{\{\mathcal{E}_{mc}\}}\right]\right| \leq \frac{1}{p^{\mathcal{P}}}\sqrt{\frac{2\log(16/\delta)}{|\mathcal{X}_A|}}.$$
(39)

Using Hoeffding's inequality we can further bound the RHS of (38) and (39). For the set $\mathcal{X}_D$, we have with probability at least $1 - \delta/2$

$$\left||\mathcal{X}_D| - n\cdot(1-p^{\mathcal{Q}})\right| \leq \sqrt{\frac{\log(4/\delta)}{2n}},$$
(40)

and for the set $\mathcal{X}_A$, we have with probability at least $1 - \delta/2$

$$\left||\mathcal{X}_A| - n\cdot p^{\mathcal{Q}}\right| \leq \sqrt{\frac{\log(4/\delta)}{2n}}.$$
(41)

We next bound the difference between the finite sample QTC estimation on the source from its expectation. By DKWM inequality, for any $q > 0$ we have with probability at least $1 - \delta/4$

$$\left|\frac{1}{|\mathcal{D}_{cal}^{\mathcal{P}}|}\sum_{\mathbf{x}\in\mathcal{D}_{cal}^{\mathcal{P}}}\mathbb{1}_{\left\{\max_{j\in\{0,1\}} f_j(\mathbf{x})<q\right\}} - \mathbb{E}_{(\mathbf{x},y)\sim\mathcal{P}}\left[\mathbb{1}_{\left\{\max_{j\in\{0,1\}} f_j(\mathbf{x})<q\right\}}\right]\right| \leq \sqrt{\frac{\log(8/\delta)}{2n}}.$$
(42)

We first show the result for the case $w_{sp} > 0$. Combining equations (38) and (42) using the triangle inequality and union bound, we have with probability at least $1 - \delta/2$

$$\left|\frac{1}{|\mathcal{D}_{cal}^{\mathcal{P}}|}\sum_{\mathbf{x}\in\mathcal{D}_{cal}^{\mathcal{P}}}\mathbb{1}_{\left\{\max_{j\in\{0,1\}} f_j(\mathbf{x})<q\right\}} - \mathbb{E}_{(\mathbf{x},y)\sim\mathcal{P}}\left[\mathbb{1}_{\{\mathcal{E}_{mc}\}}\right]\right| \leq \frac{1}{1-p^{\mathcal{P}}}\sqrt{\frac{2\log(16/\delta)}{|\mathcal{X}_D|}}.$$
(43)

Plugging in the definitions of $\beta_{\mathrm{QTC}}$ in (19) and $\beta$ in (16) above, equivalently we get

$$|\beta_{\mathrm{QTC}} - \beta| \leq \frac{1}{1-p^{\mathcal{P}}}\sqrt{\frac{2\log(16/\delta)}{|\mathcal{X}_D|}},$$
(44)

which holds with probability at least $1 - \delta/2$. Combining (44) with (40) proves equation (20) for $w_{sp} > 0$, as desired.

Similarly, for the case $w_{sp} < 0$, following the same steps by first combining equation (39) with (42), we have with probability at least $1 - \delta/2$

$$|\beta_{\mathrm{QTC}} - \beta| \leq \frac{1}{p^{\mathcal{P}}}\sqrt{\frac{2\log(16/\delta)}{|\mathcal{X}_A|}}.$$
(45)

Combining (45) with (41) yields equation (20), as desired, for the case $w_{sp} < 0$, which concludes the proof.

$\square$

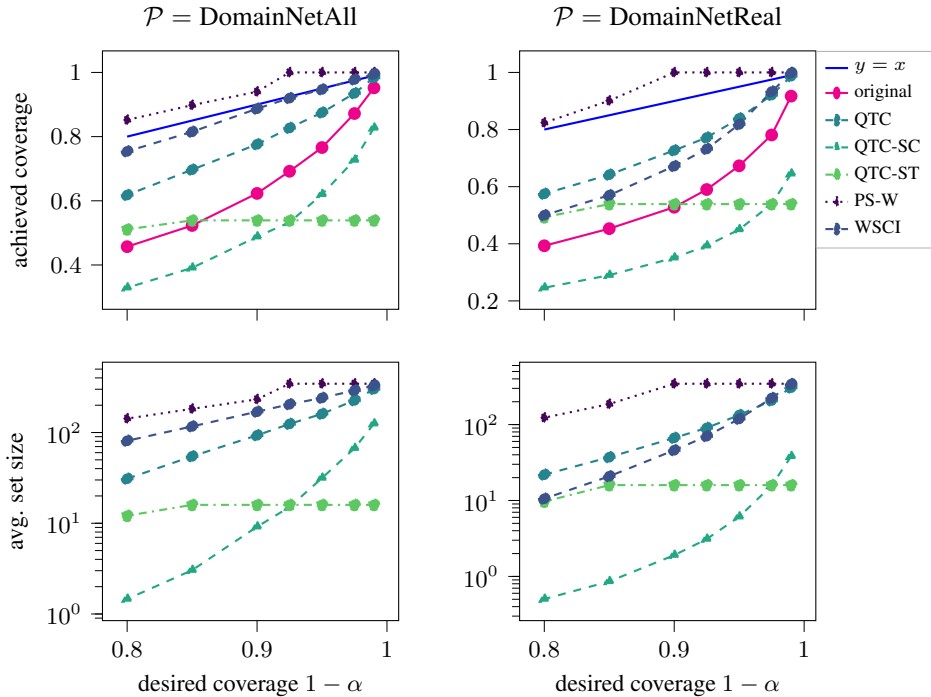

Figure 5: Coverage (**top row**) and the average set size (**bottom row**) obtained by TPS on the target distribution $\mathcal{Q} = $ DomainNet-Infograph for various settings of $(1 - \alpha)$ after recalibration with QTC (ours) and covariate shift based methods of WSCI (Tibshirani et al., 2019) and PS-W (Park et al., 2022). We show the results for the setting where all domains (including the target) are available as the source (**left**) as well as for the setting where all domains are available at training time but only the DomainNet-Real domain is available for the source at calibration time (**right**). For the setting of $\mathcal{P} = $ DomainNetAll, WSCI closes the coverage gap while QTC considerably improves it, whereas for the setting of $\mathcal{P} = $ DomainNetReal, QTC slightly outperforms WSCI. In both settings, PS-W fails by constructing uninformatively large confidence sets for the range $1 - \alpha > 0.9$ and QTC-SC and QTC-ST fail to successfully recalibrate the conformal predictor.

# B    COMPARISON TO METHODS BASED ON ESTIMATING A COVARIATE SHIFT

In this section, we compare the performance of QTC to methods that assume a covariate shift as a distribution shift, and recalibrate by estimating the amount of covariate shift from unlabeled target data (Tibshirani et al., 2019; Park et al., 2022).

Covariate shift refers to the case where the conditional distribution of the label $y$ given the feature vector $\mathbf{x}$ is fixed across the source and target distributions, but the marginal distribution of the feature vectors differ. Specifically, we have

$$\begin{aligned} \text{source} \quad &: \quad (\mathbf{x}, y) \sim \mathcal{P} = p_{\mathcal{P}}(\mathbf{x}) \times p(y|\mathbf{x}) \\ \text{target} \quad &: \quad (\mathbf{x}, y) \sim \mathcal{Q} = p_{\mathcal{Q}}(\mathbf{x}) \times p(y|\mathbf{x}), \end{aligned}$$

where $p_{\mathcal{P}}(\mathbf{x})$ and $p_{\mathcal{Q}}(\mathbf{x})$ are the marginal PDFs of the feature vector $\mathbf{x}$, $p(y|\mathbf{x})$ is the conditional PDF of the label $y$, and $p_{\mathcal{P}}(\mathbf{x}) \neq p_{\mathcal{Q}}(\mathbf{x})$.

In order to account for the covariate shift, the methods from (Tibshirani et al., 2019; Park et al., 2022) utilize an approach called *weighted conformal calibration*. In weighted conformal calibration, the likelihood ratio of the covariate distributions, i.e., $w(\mathbf{x}) = p_{\mathcal{Q}}(\mathbf{x})/p_{\mathcal{P}}(\mathbf{x})$, is used to weigh the scores used for the set generating function of the conformal predictor for each sample $(\mathbf{x}, y) \in \mathcal{D}_{\text{cal}}^{\mathcal{P}}$.

The weights $w(\mathbf{x})$ are called the *importance weights* (IWs) and the conformal predictor calibrated on a source calibration set with the true importance weights (if known) for a given target distribution

is guaranteed to achieve the desired coverage on the target, see Tibshirani et al. (2019, Cor. 1). In practice, the true IWs are not known and are therefore estimated heuristically.

In general, covariate shift is not well defined for complex tasks such as image classification. We therefore follow the experimental setup of Park et al. (2022) and consider a backbone ResNet-101 classifier trained using unsupervised domain adaptation based on training sets sampled from both the source and target distribution. Furthermore, following Park et al. (2022), we train an auxillary classifier (discriminator) $g$ that yields probability estimates of a given input example being drawn from the source distribution or the target distribution. For the *weighted split conformal inference* (WSCI) method of Tibshirani et al. (2019), we estimate the importance weights using this discriminator $g$. For the PAC prediction sets method of Park et al. (2022) based on rejection sampling (PS-W), we estimate the importance weights using histogram density estimation over the output probabilities of the discriminator $g$ on the unlabeled examples of the source and target calibration sets. We use TPS as the conformal predictor for all of the considered methods. For details on the experimental setup for the covariate shift based methods, see Park et al. (2022, Sec. 4.1)

Note that QTC is significantly more versatile compared to the above covariate shift based methods in that QTC applies in the setting where labeled examples from the target distribution may not be available during training to train an auxillary source-target discriminator, which is required to estimate the importance weights for the covariate shift based methods (Tibshirani et al., 2019; Park et al., 2022).

In order to compare QTC to WSCI and PS-W, following Park et al. (2022), we consider the DomainNet distribution shift problem (Peng et al., 2019). We first consider the original setup of Park et al. (2022), where the source distribution comprises of multiple domains and the target distribution corresponds to only specific domain out of these multiple domains. Specifically, the source distribution contains examples from all six domains of the DomainNet dataset, which are *DomainNet-Sketch, DomainNet-Clipart, DomainNet-Painting, DomainNet-Quickdraw, DomainNet-Real, and DomainNet-Infograph*. In contrast, the target distribution contains examples from only the *DomainNet-Infograph* domain. We focus on the DomainNet-Infograph domain for the target distribution as the performance drop for other choices of the target domains are considerably small or even negligible even without any recalibration (see Park et al. (2022, Table 1)). We also consider a more challenging scenario in which all domains are available at training time, but only a single domain (different than the target) is available for the source at calibration time. For this setting, inspired by the ImageNet distribution shift setups, we consider the DomainNet-Real domain that features real images as the source and the DomainNet-Infograph domain as the target.

We present our findings in Figure 5. The results show that when the source includes all the domains, WSCI outperforms other methods. However, when only DomainNet-Real is available for the source at calibration time, QTC slightly outperforms WSCI. In both settings, PS-W fails if $\alpha$ is chosen such that $1 - \alpha > 0.9$, by constructing uninformatively large confidence sets that tend to contain all possible labels. On the other hand, QTC and WSCI tend to construct similarly sized confidence sets consistently across the range of $1 - \alpha$. Note that while QTC considerably closes the coverage gap in both setups, QTC-SC and QTC-ST fail to improve the coverage gap. We believe this might be due to the fact that ResNet-101 trained with domain adaptation tends to yield very high confidence across all examples. While a separate discriminator that uses the representations of the ResNet-101 before the fully-connected linear layer is utilized for the covariate shift based methods, this is not the case for QTC and its variants. Therefore, the threshold found by QTC-SC and QTC-ST tends to be very close or even equal to $1.0$, hindering the performance.

## C  RAPS RECALIBRATION EXPERIMENTS

APS is a powerful yet simple conformal predictor. However, other conformal predictors (Sadinle et al., 2019; Angelopoulos et al., 2020) are more efficient (in that they have on average smaller confidence sets for a given desired coverage $1 - \alpha$).

In this section, we focus on the conformal predictor proposed by Angelopoulos et al. (2020), dubbed Regularized Adaptive Prediction Sets (RAPS). RAPS is an extension of APS that is obtained by adding a regularizing term to the classifier's probability estimates of the higher-order predictions (i.e., subsequent predictions after the top-k predictions). RAPS is more efficient and tends to produce

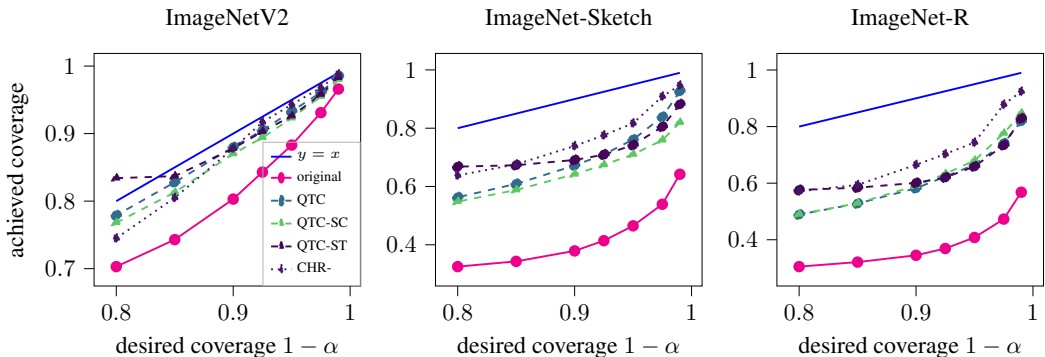

Figure 6: Coverage obtained by RAPS on the target distribution $\mathcal{Q}$ for various settings of $(1 - \alpha)$ w/ and w/o recalibration using QTC.

smaller confidence sets when calibrated on the same calibration set as APS, as it penalizes large sets. While TPS tends to achieve slightly better results in terms of efficiency compared to RAPS, see (Angelopoulos et al., 2020, Table 9), RAPS coverage tends to be more uniform across different instances (in terms of difficult vs. easy instances) and therefore RAPS still carries practical relevance.

Recall that while efficiency can be improved by constructing confidence sets more aggressively, efficient models tend to be less robust, meaning the coverage gap is greater when there is distribution shift at test time. For example, when calibrated to yield $1 - \alpha = 0.9$ coverage on ImageNet-Val and tested on Image-Sketch, the coverage of RAPS drops to $0.38$ in contrast to that of APS, which only drops to $0.64$ (see Section 2). It is therefore of great interest to understand how QTC performs for recalibration of RAPS under distribution shift.

RAPS is calibrated using exactly the same conformal calibration process as APS and only differs from APS in terms of the prediction set function $\mathcal{C}(\mathbf{x}, u, \tau)$. The prediction set function for RAPS is defined as

$$\mathcal{C}^{\text{RAPS}}(\mathbf{x}, u, \tau) = \left\{ \ell \in \{1, \ldots, L\} \colon \sum_{j=1}^{\ell-1} [\pi_{(j)}(\mathbf{x}) + \underbrace{\mathbb{1}_{\{j - k_{\text{reg}} > 0\}} \cdot \lambda}_{\text{regularization}}] + u \cdot \pi_{(\ell)}(\mathbf{x}) \leq \tau \right\}, \quad (46)$$

where $u \sim U(0, 1)$, similar to APS and $\lambda, k_{\text{reg}}$ are the hyperparameters of RAPS corresponding to the regularization amount and the number of top non-penalized predictions respectively.

Note that the cutoff threshold $\tau^{\mathcal{P}}$ obtained by calibrating RAPS on some calibration set $\mathcal{D}_{\text{cal}}^{\mathcal{P}}$ can be larger than one due to the added regularization. Therefore, in order to apply QTC-ST, we map $\tau^{\mathcal{P}}$ back to the $[0, 1]$ range by dividing by the total scores after added regularization. QTC and QTC-SC do not require such an additional step as the coverage level $\alpha \in [0, 1]$ by definition. We show the results of RAPS' performance under distribution shift with or without calibration by QTC in Figure 6. The results show that while QTC is not able to completely mitigate the coverage gap, it significantly reduces it.

Recall that RAPS utilizes a hyperparameter $\lambda$, which is the added penalty to the scores of the predictions following the top-$k_{\text{reg}}$ predictions, that can significantly change the cutoff threshold $\tau^{\mathcal{P}}$ when we calibrate on the calibration set $\mathcal{D}^{\mathcal{P}}$. The regularization amount $\lambda$ also implicitly controls the change in the cutoff threshold $\left| \tau^{\mathcal{Q}} - \tau^{\mathcal{P}} \right|$ when the conformal predictor is calibrated on different distributions $\mathcal{P}$ and $\mathcal{Q}$. That is, the value of $\left| \tau^{\mathcal{Q}} - \tau^{\mathcal{P}} \right|$ increases with increasing $\lambda$ as long as the distributions $\mathcal{P}$ and $\mathcal{Q}$ are meaningfully different, as is the case for all the distribution shifts that we consider.

Therefore, a good recalibration method should be relatively immune to the choice of $\lambda$ in order to successfully predict the threshold $\tau^{\mathcal{Q}}$ based only on unlabeled examples. In Figure 7, we show the performance of RAPS under the ImageNetV2 distribution shift for various values of $\lambda$. While QTC is able to improve the coverage gap for various choices of $\lambda$, the best performing regression-based baseline method does not generalize well to natural distribution shifts when $\lambda$ is relatively large. In

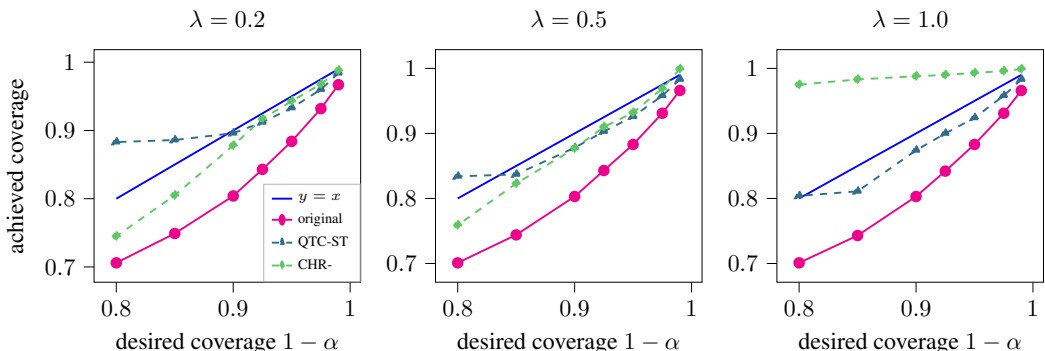

Figure 7: Coverage obtained by RAPS on the target distribution $\mathcal{Q}$ (ImageNetV2) for $k_{\mathrm{reg}} = 2$ and various settings of $\lambda$ when the threshold $\tau$ is replaced with the predicted threshold $\hat{\tau}$ with the respective prediction method. For regression methods, only the best performing method of CHR- is shown.

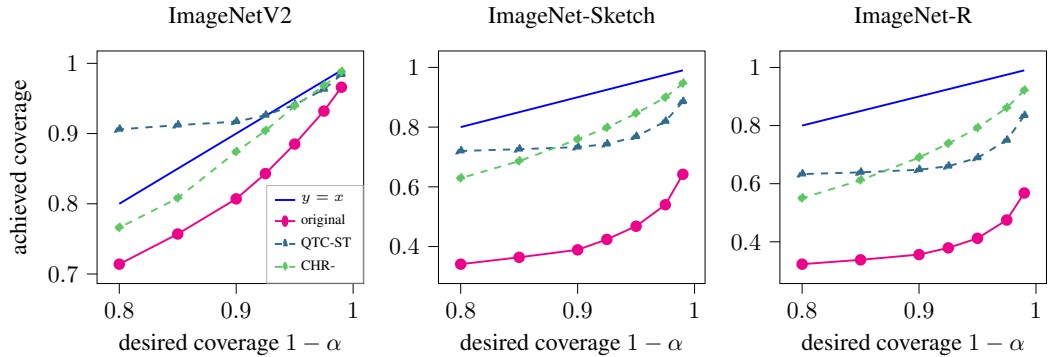

Figure 8: Coverage obtained by RAPS on the target distribution $\mathcal{Q}$ for $\lambda = 0.1$ and various settings of $(1 - \alpha)$ when the threshold $\tau$ is replaced with the predicted threshold $\hat{\tau}$ with the respective prediction method. For regression methods, only the best performing method of CHR- is shown.

contrast, as demonstrated in Figure 8, when the regularization amount $\lambda$ is relatively small, the best performing regression-based method of CHR- does very well in reducing the coverage gap of RAPS under various distribution shifts.

