# OpenReview forum: "Test-time recalibration of conformal predictors under distribution shift based on unlabeled examples"
_ICLR.cc/2023/Conference — Submitted to ICLR 2023_

### Official Review · Reviewer_4zCm · 2022-10-24

**Confidence:** 4
**Clarity, Quality, Novelty And Reproducibility:** See above.
**Correctness:** 3
**Technical Novelty And Significance:** 3
**Empirical Novelty And Significance:** 2
**Recommendation:** 5

**Strength And Weaknesses:**

Strengths:
- Thorough introduction and problem statement and nice review of work on accuracy estimation with distribution shift and conformal prediction with distribution shift.
- The authors tackle two important problems in my opinion: First, recalibration under distribution shift, second, calibration with unlabeled data.
- Realistic experiments on ImageNet.
- Theoretical analysis based on prior work that may give some indication why the method works.

Weaknesses:
- The paper is a bit notation heavy due to the strong reliance on thresholds, confidence levels and quantiles and I feel giving a bit more intuition on the calibration processes could help the reader.
- Notation-wise, the opposite inequalities in (3) and (4) are confusing and I do not understand why 1 – alpha is used in (7). Generally, of course, there is some duality (e.g., replacing the 1 – tau with tau in equation (3) will correspond to calibrating against alpha instead of 1 – alpha), but it seems that this is not consistent across equations.
- Figure 1 is aimed at illustrating the method, but it does not work well for me. First, the steps in the caption where confusing at first (are these text or part of the caption) and context for the confidence levels and thresholds is missing, i.e. the figure is on its own not helpful. Also, I am missing the connection to the calibration with TPS/APS with labels.
- Some related work is missing in my opinion, e.g., online adaptive calibrators such as [a] that I feel are very similar.

[a] https://arxiv.org/pdf/2106.00170.pdf

- For the method, I think that I am missing an important point. The calibration on P is done with labels, correct? Meaning, we calibrate the threshold tau^P such that coverage 1 – alpha on P is obtained and the APS/TPS scores are used for that. However, there is no intuition or connection whatsoever between calibration of APS/TPS with labels and the unlabeled score s on P. For APS, the score is not even similar (while for TPS, s is just the maximum conformity score). This seems to be an integral part because calibrating against alpha on P using s might not mean anything for the actual TPS or APS scores. This is of course because the maximum confidence might not correspond with the true label.
- I am also slightly confused by the QTC-ST baseline. For the conformity score being the maximum confidence, this works. But if my scores are logits, equation (5) does not seem to work since tau^P could be outside of [0, 1] while the confidence levels are in [0, 1]. For tau^P > 1, equation (5) should not work/be meaningful.
- In section 3.2, it is also extremely difficult to follow why we are suddenly interested in regression baselines that predict accuracy. I expect that these are used to predict coverage from which beta is derived? The experiments section does, unfortunately not provide any more details.
- I am also missing a clear statement that this approach does not provide any guarantee. Often, coverage is associated with a guarantee and I think this should be discussed and stated accordingly.
- For the theoretical analysis, I would expect a proper statement in Theorem 1 beyond just saying “converges to 1 – alpha with high probability”. I feel without an exact statement, the toy example is not insightful. From the appendix, I deduce that this does not correspond to a standard coverage guarantee but, but a guarantee on that beta can be estimated reasonable close. However, only because beta is close, coverage does not need to be close, too. Or am I missing anything?

**Summary Of The Paper:**

The authors propose a method for recalibrating a conformal predictor on a shifted data distribution with only unlabeled data. To this extent, they intend to estimate a confidence level beta that, if used for calibration on the original data distribution, would give empirical coverage 1 – alpha (the target coverage) on the target distribution. This is done using a unlabeled confidence-score.

**Summary Of The Review:**

In its current form I am not fully convinced that this paper is ready for publication. There are several disconnects that are missing for me to fully understand the details and judge the method properly.

---

> ### Author Response · Authors · 2022-11-16
> **Authors' Response to Reviewer 4zCm**
>
> We thank the reviewer for the valuable feedback and noting the significance of the problem. We address the reviewer’s feedback and questions below:
>
> - Regarding the notations in eq. (3) and (4), we used $1-\tau$ for TPS and $\tau$ for APS intentionally to capture the nesting property of the generated sets. Since TPS thresholds each score independently, we use $1-\tau$ in eq. (3) such that increasing $\tau$ leads to larger, nesting sets. Thank you for noting about the confusion, we are revising the presentation of the methods and equations across the paper to avoid future confusions.
> - Regarding the missing related work, we initially limited the mention of related work in the context of conformal prediction to the papers that propose distribution shift robust methods without using additional labeled data and hence omitted the adaptive or rolling conformal inference methods as they rely on continuously available labeled data. Thanks for bringing them to our attention, we now also cite the latter works.
> - Regarding the motivation behind using the model confidence to estimate the APS/TPS conformal thresholds, we note that model confidence has been used extensively in the past particularly in the fields of classifier accuracy prediction and out-of-distribution (OOD) detection (Lee et al., 2018)[^1] with great success. This, together with the intuition from our theoretical results, provide the motivation for estimating the target coverage level based on the scores of the unlabeled samples. To give a simple example, for a well-calibrated model, we would expect harder problem instances/samples to not only have lower prediction accuracy, but also lower confidence. Therefore, for a well-calibrated model, one can estimate a metric that depends on the labels by using the scores for unlabeled examples. Even though models are not very well calibrated in practice, our theoretical and empirical results provide a strong intuition for using the scores to predict the coverage level of a conformal predictor, especially TPS which individually depends on the model scores.
> - Regarding the expectation that QTC-ST should not work for the settings where the threshold $\tau$ can be larger than one, this is indeed true and is addressed in the paragraph following eq. (46). Specifically, we map the threshold $\tau$ back to the $(0,1)$ range in order to apply QTC-ST. We provide QTC-ST only as a baseline version of QTC that directly estimates the threshold $\tau$ similar to the regression based baselines. The lackluster performance of the QTC-ST, together with the above shortcoming makes it a nonviable option. We focus mainly on QTC and will move the other variations, including QTC-ST, to the supplementary material for the sake of clarity.
> - Regarding why we are interested in regression based baselines: Since regression based methods have been used successfully to predict the accuracy, we consider similar methods for predicting the conformal threshold $\tau$, to have baselines. We use both existing features that have been state-of-the-art at some point (for the out of scope task of predicting the accuracy) as well as additional features extracted from the model-dataset statistics that we test. We added clarifications in Section 3.2 to address the confusion.
> - Regarding the lack of a clear statement on the lack of guarantees, we state that it is generally impossible to guarantee coverage and that we rely on the results instead both in the last sentence of the abstract as well as the second to the last paragraph before Section 3. If the wording is not clear, we are happy to rephrase.
> - Regarding the statement of Theorem 1, while our direct result is that $\beta$ is guaranteed to be close, for real world datasets where the distribution of the model confidence is relatively smooth across the dataset samples, this translates to coverage being close. This is due to the nesting property of the confidence sets of the conformal predictors that we consider.
>
> &nbsp;
>
> [^1]: Lee, Kimin, Kibok Lee, Honglak Lee, and Jinwoo Shin. A simple unified framework for detecting out-of-distribution samples and adversarial attacks. In *Advances in Neural Information
> Processing Systems (NeurIPS)*. 2018.

---

### Official Review · Reviewer_h3g1 · 2022-10-25

**Confidence:** 3
**Clarity, Quality, Novelty And Reproducibility:** The paper is well organized, and the …
**Correctness:** 3
**Technical Novelty And Significance:** 2
**Empirical Novelty And Significance:** 2
**Recommendation:** 3

**Strength And Weaknesses:**

Strength:
Appropriate experiments are conducted with some comparative methods.

Weaknesses:
-- The coverage is not exact. Actually, the proposed method is even not guaranteed to achieve the desired coverage probability asymptotically, which largely limits the contribution and novelty of this paper.

-- I am curious about efficiency about the recalibration method. This paper does not provide enough analysis and experimental results about efficiency. Maybe the authors can provide some experiments to compare the size of the predictive sets of different methods (on synthetic or real data sets). Simultaneous analysis of validity and efficiency is more informative and meaningful.

-- Comparisions in more datasets with conformal methods under distribution shift are recommended. The current version only provides comparison with some covariate-shift-based conformal methods in Appendix B. From the results (e.g., Figure 5), the performance of the proposed method is not convincing.

-- It is not clear which one of the three recalibration methods we should use in practice. From experimental results, each one may fail to achieve the desired coverage on target data (the gap may be large). Can the authors provide more discussions and practical guidance?

-- Why in Figures 7 and 8, are only results of QTC-ST provided? From other results, QTC-ST seems more likely to perform worse than the other two recalibration methods. Please provide all relevant results.

-- The authors only provide some theoretical results under a toy example. For some other types of distribution shift such as covariate shift (unlabeled data seems sufficient), can the proposed method guarantee the coverage on the target data?

-- What is the main difference between the proofs in this paper and that in Garg et al. (2022)? Please state clearly about the additional effort required in this work.

**Summary Of The Paper:**

Conformal prediction methods guarantee validity only when the the calibration set comes from the same distribution as the test set. This paper aims to construct a more reliable conformal predictive set under distribution shift with unlabeled data. They consider predicting the cutoff threshold for a new distribution based on unlabeled data only. Experiments are conducted to demonstrate the performance of the proposed method in reducing the coverage gap.


**Summary Of The Review:**

Though the paper proposes a simple method to obtain a predictive set under distribution shift, the experimental results are not very convincing. Moreover, the coverage is not guaranteed, which is a big issue.

---

> ### Author Response · Authors · 2022-11-16
> **Authors' Response to Reviewer h3g1**
>
> We thank the reviewer for the valuable feedback. We address the reviewer’s feedback and questions below:
>
> - Regarding the proposed method not having coverage guarantees, we remark that the existing methods for recalibration or robust calibration of conformal predictors only provide guaranteed coverage under strong assumptions, such as only considering covariate shifts. Similarly, we show that our method is guaranteed to achieve the desired coverage for a concrete distribution shift setting by Nagarajan et al. (2021); Garg et al. (2022). For natural, complex distribution shifts coverage guarantees do not exist. In fact, covariate shift based methods also rely on some level of prior knowledge about the source and target distributions (e.g. a pre-trained discriminator model) in order to estimate the importance weights heuristically for complex tasks such as image classification.
> - Regarding how the average size of the prediction sets compares between the proposed method and the baselines, we note that all of the considered methods aim to correctly estimate the threshold $\tau$ for a given conformal predictor, but do not otherwise alter the set generating function. Therefore, if any two methods estimate the same threshold for a given conformal predictor, the average set sizes would be equal. However, the set sizes at the same coverage level change significantly across different conformal predictors. Because of this, QTC is particularly useful as it aims to recalibrate efficient conformal predictors like TPS, which suffer significantly under distribution shift, while not compromising the efficiency. We will add a more detailed comparison for the average set sizes in the supplementary material to convey this more clearly.
> - Regarding the comparison to the covariate shift based methods, we are working on adding results for ImageNet distribution shifts in addition. On the other hand, we remark that QTC, while not outperforming all existing covariate shift based methods, compares favorably in the case where no prior knowledge of the target distribution is available to pre-train an auxiliary model, which is required for the existing methods.
> - Regarding which version of the QTC to use in practice; this is a good point. QTC is the main method of choice, we'll move the others as ablation studies to the appendix.
> - Regarding why only QTC-ST results are provided in Figures 7-8 concerning the effects of the RAPS hyperparameter choices, we note that only QTC-ST is majorly affected by different choices of RAPS hyperparameters. This is because QTC-ST directly estimates the threshold $\tau$, which can exceed the quantile range $(0,1)$ depending on $\lambda$ and $k_{reg}$. QTC and QTC-SC estimate the coverage level $\beta$, which is guaranteed to be in the range $(0,1)$.
> - Regarding the difference between the proofs in our paper and that in Garg et al. (2022), we adapt the idea/approach in Garg et al. (2022) since we consider the same distribution shift, but we show results for a different task. The proofs are different and the results in Garg et al. (2022) do not imply our results.

---

### Official Review · Reviewer_gDoT · 2022-10-28

**Confidence:** 3
**Correctness:** 4
**Technical Novelty And Significance:** 2
**Empirical Novelty And Significance:** 2
**Recommendation:** 5

**Clarity, Quality, Novelty And Reproducibility:**

The presented method is simple both in terms of explanation and in terms of use, though some improvements (example) are needed (see above). The step of using only unlabeled data to calibrate conformal prediction is interesting, though its robustness is questionable. Perhaps I would suggest plotting similar figures for smaller datasets such as the cifar-10/100 to strengthen the results, namely whether the same positive effect of recalibration on a smaller amount of data would occur.

**Strength And Weaknesses:**

Strengths:
* The work is neatly written and easy to read.
* The problem presented is relevant, which is confirmed by many examples of the lack of guarantees when the test distribution is shifted compared to the calibration set.
* For the experimental study, authors use variations of ImageNet, which is a large and challenging dataset, which makes the obtained results rather strong.
* The paper also presents a comparison with a number of baselines and an extensive theoretical study of the proposed method.

Weaknesses:
* While reading is easy the main idea of predicting the quantile is not explained enough. To the best of my understanding, you characterize datasets by some features and try to predict the quantiles with some predictor based on this features. It is not that easy to grasp from the paper. I recommend giving some example in the main body of the paper.
* Are there any theoretical guarantees for the resulting method? Why do we expect it to work well?
* While experimental results are promising, I am really curious how stable is the solution. It worked well on give example but what will predictor output for some very new data (with more significant shift for example).
* The method is a direct adaptation of the analysis of Garg et al. 2022, so the contribution is limited.

**Summary Of The Paper:**

The authors approach the recalibration of the prediction set in the conformal prediction procedure to overcome the distribution shift using only unlabeled data. Although this problem is intractable in general terms, it can be solved using the assumption about the family of shifts used. The authors propose a simple method Quantile Thresholded Confidence (QTC) which proposes to estimate the quantile on the target distribution on the desired level, after which a parameter beta is calculated to calibrate the predictor on the calibration data. To calculate threshold on source data, it is proposed either to search for parameter beta and estimate threshold from it, or to estimate threshold directly.

**Summary Of The Review:**

The paper is well written and the results presented are interesting, but I have serious doubts on robustness of the method. No theoretical guarantees are given which is an important part of success for conformal prediction.

---

> ### Author Response · Authors · 2022-11-16
> **Authors' Response to Reviewer gDoT**
>
> We thank the reviewer for the valuable feedback and noting the significance of the problem and the appeal of our approach. We address the reviewer’s feedback and questions below:
>
> - Regarding better clarifying the main idea of predicting the quantile: Thanks for pointing this out, we'll revise the paper to do a better job to explain the main idea. Our approach to explain the idea is to consider two events:
>     1. The event that a sample in the dataset is not covered by the conformal predictor (i.e. true label not included in the confidence set).
>     2. The event that the model confidence is less than some computed threshold for a sample.
>
>     Let us denote by Set-1 the set of samples for which the first event holds, and Set-2 the set of samples for which the second event holds. Our intuition is that while the sizes of these sets change under distribution shift, the relative change across the sets is similar, which is made concrete by our theoretical results for the toy problem that we consider. Taking the quantile simply enables us to initially match the size of the Set-2 to that of Set-1 by computing the appropriate threshold for convenience. Note that, ultimately, the goal of recalibration can be reduced to estimating the size of Set-1 which enables the success of our method.
> - Regarding the intuition and why we expect our method to work well: Prior works have demonstrated that the model-dataset statistics such as the maximum model confidence can be used to predict the accuracy of the classifier with great success. We show that, on the same toy model that motivates the state-of-the-art accuracy prediction method (Garg et al., 2022), the conformal predictor is guaranteed to achieve the desired $1-\alpha$ coverage after recalibration with QTC. Extensive empirical success on the task of accuracy prediction as well as our extension to the theory and empirical evidence motivates our belief that recalibration by QTC is expected to work well.
> It is in general not possible to provide guarantees for practical distribution shifts without imposing strong assumptions, such as considering covariate shifts. Even then, the existing covariate shift based methods rely on the existence of a pre-trained discriminator that has prior knowledge of both the source and target distributions for practical applications (Park et al., 2022).
> - Regarding the stability of the recalibration solution, most of the distribution shifts that we consider in our experimental setup pose a severe shift. For example, the ImageNet-Sketch distribution, which contains only sketches of objects, causes the actual coverage to drop to 38% from the target coverage of $1-\alpha=0.9$ for TPS. Similarly, The BREEDS shifts pose a drop of 30-45% in coverage. If we imagine a worst case scenario where the target distribution comprises many out-of-distribution (OOD) classes with respect to the training/calibration set, our solution would not work well in this setting (without fine-tuning the underlying classifier). However, we believe this setting is outside the scope of our work and is better suited for the vast OOD-detection literature, which can always be employed to preprocess the target data to eliminate the outliers.

---

> > ### Comment · Reviewer_gDoT · 2022-11-17
> > **Thanks for the comments**
> >
> > Dear authors,
> >
> > I appreciate your comments that improved my understanding of the paper and the results. I decided to raise my score but I still think that the present approach lacks the strict guarantees of conformal prediction. It would be of interest to see the validity of the approach on some broad class of shifts and/or models. The present theory is written for the very particular model and data distribution that well match together. Thus, it is not clear why generally the proposed recalibration procedure should work well. I recommend to extend experimental evaluation if to show the applicability of the method to the wide set of models and datasets.

---

### Official Review · Reviewer_9GdY · 2022-11-03

**Confidence:** 4
**Clarity, Quality, Novelty And Reproducibility:** Clear. See comments above.
**Correctness:** 2
**Technical Novelty And Significance:** 2
**Empirical Novelty And Significance:** 2
**Recommendation:** 3

**Strength And Weaknesses:**

Strength: this paper is clearly written. Most parts are easy to follow.

Weakness:

1. The method proposed is not intuitive, I really double whether this method will work on multiple class classification problems theoretically. From the expression of 6, it seems it only based on the maximal logic, and unlike TPS and APS, based on all logic explicitly. A theoretical study beyond 2 classes is appreciated.

2. The experiments cannot justify a wide family of natural distribution shifts. More datasets should be discussed to show more comprehensive results.

In summary, the beauty of distribution-free uncertainty quantification is rigorous analysis and handy to use, but this paper missed these parts.

**Summary Of The Paper:**

This paper considers the problem of domain shift at test time. They consider how to use the shifted unlabeled test data to recalibrate the cutoff threshold. The method is shown to be valid for some natural distribution shifts. Their method is mainly based on QTC in equation 6. Validity is also shown in binary classification in section 5. Empirical methods are based on image net and breeds.

**Summary Of The Review:**

See comments above.

---

> ### Author Response · Authors · 2022-11-16
> **Authors' Response to Reviewer 9GdY**
>
> We thank the reviewer for the feedback. We address the reviewer's feedback and questions below:
>
> - Regarding the intuition of our method and the lack of theory for multiple classes: The intuition of our method is that changes in the predicted probabilities of a model can be reflective of a distribution shift, and can be used to re-calibrate. This intuition is supported by a theoretical result for two classes for one particular distribution shift introduced by Nagarajan et al. (2021) and studied by  Garg et al. (2022) in the context of predicting model accuracy. The method is expected to work on some distribution shifts (like the one by Garg et al. (2022)), but is not expected to work universally. We demonstrated that it works well for a few multi-class shifts, but unfortunately do not have theoretical examples of shifts for which it provably works for a multi-class classification problem.
> - Regarding the scope of the experiments, we consider many of the natural distribution shifts (datasets) that are used as standard benchmarks across the literature. As we are primarily concerned with potential real world use cases, we refrain from testing on synthetic distribution shifts, which often comprise the majority of the shifts used. If you have a particular additional shift in mind, we're happy to study it for a revised version of the paper.

---

### Decision · Program_Chairs · 2023-01-20

**Decision:**

Reject

**Justification For Why Not Higher Score:**

N/A

**Justification For Why Not Lower Score:**

N/A

**Metareview: Summary, Strengths And Weaknesses:**

This paper considers the problem of uncertainty quantification (UQ) of predictors using the conformal prediction framework when there is a change in the data distribution (source => target) using only unlabaled examples from the target distribution. This is a very important problem.

It proposes a method referred to as Quantile Thresholded Confidence (QTC) which estimates a confidence level beta that, if used for calibration on the source distribution, would give empirical target coverage 1 – alpha on the target distribution. This estimation is performed using a unlabeled confidence-score. QTC is inspired by and is an adaptation of Garg et al., 2022 which studied the problem of predicting out-of-distribution performance using unlabaled data. Overall, the paper is very well-written and ideas are easy to follow.

There is a consensus among the reviewers' that the paper is lacking in the following aspects and requires some more work before it is ready for publication.
1. The novelty of the QTC method is limited as it is an adapatation of Garg et al., 2022 for the conformal prediction setting.
2. The key advantage of conformal prediction methods over prior methods for UQ is strong theoretical guarantees. There is some theoretical analysis on a toy setup, but lacks strong and general theoretical analysis.
3. Experimental evaluation could be more comprehensive by testing on a wide range of models and distribution shifts to understand the robustness of the proposed recalibration method.

In my mind, #1 is not a major concern if #2 and/or #3 was addressed sufficiently well. I'm recommending to reject this paper and strongly encourage the authors' to address at least one out of #3 and #2 for resubmission.